# Rif1 restrains the rate of replication origin firing in *Xenopus laevis*

Olivier Haccard [1], Diletta Ciardo [2], Hemalatha Narrissamprakash[1], Odile Bronchain [3], Akiko Kumagai [4], William G. Dunphy[4], Arach Goldar[1] & Kathrin Marheineke [1✉]

Metazoan genomes are duplicated by the coordinated activation of clusters of replication origins at different times during S phase, but the underlying mechanisms of this temporal program remain unclear during early development. Rif1, a key replication timing factor, inhibits origin firing by recruiting protein phosphatase 1 (PP1) to chromatin counteracting S phase kinases. We have previously described that Rif1 depletion accelerates early *Xenopus laevis* embryonic cell cycles. Here, we find that in the absence of Rif1, patterns of replication foci change along with the acceleration of replication cluster activation. However, initiations increase only moderately inside active clusters. Our numerical simulations suggest that the absence of Rif1 compresses the temporal program towards more homogeneity and increases the availability of limiting initiation factors. We experimentally demonstrate that Rif1 depletion increases the chromatin-binding of the S phase kinase Cdc7/Drf1, the firing factors Treslin, MTBP, Cdc45, RecQL4, and the phosphorylation of both Treslin and MTBP. We show that Rif1 globally, but not locally, restrains the replication program in early embryos, possibly by inhibiting or excluding replication factors from chromatin.

[1] Université Paris-Saclay, CEA, CNRS, Institute for Integrative Biology of the Cell (I2BC), 91198 Gif-sur-Yvette, France. [2] Institut de Biologie de l'Ecole Normale Supérieure, Ecole Normale Supérieure, CNRS, INSERM, PSL Research University, Paris, France. [3] Paris-Saclay Institute of Neuroscience, CNRS, Université Paris-Saclay, CERTO-Retina France, 91400 Saclay, France. [4] Division of Biology and Biological Engineering, California Institute of Technology, Pasadena, CA 91125, USA. ✉email: kathrin.marheineke@i2bc.paris-saclay.fr

DNA replication in eukaryotes starts at several hundred to thousands of sites called replication origins, which are activated during the S phase according to a regulated spatio-temporal replication program[1–3]; the dysregulation of this program leads to genomic instability. Genome-wide studies have shown that large adjacent genomic segments, called replication domains, share a similar replication timing (RT or mean replication time of a locus in a cell population). Early replication generally occurs in actively transcribed euchromatin in the A compartments in the nucleus of differentiated cells. During early developmental stages, when transcription has not begun and without distinct eu/heterochromatin, the rapid cell cycles rely only on maternally supplied factors; whether or how replication timing during this developmental period is regulated remains unclear. We and others previously found that in the *Xenopus* in vitro system, active replication origins are spaced 5–15 kb apart[4,5], clustered in early- and late-firing groups of origins[5,6], and that an embryonic temporal timing program exists in the *Xenopus* egg extract system and early embryos from *Xenopus* and Zebrafish[6–9].

The coordination of more than fifty different protein factors is necessary to (i) license, (ii) activate (fire) a replication origin, (iii) establish two replication forks, and (iv) achieve the faithful duplication of the genetic material. In late mitosis and G1 phase, origins are licensed for replication by loading onto chromatin the pre-replicative complex (pre-RC, for review, see[2]), composed of the six ORC (origin recognition complex) subunits, the Cdc6 (cell-division-cycle 6) and the MCM (mini-chromosome maintenance) 2–7 helicase complex. Cyclin- and Dbf4/Drf1-dependent kinases (CDKs and DDKs) activate the pre-RC at the start and during S phase. In budding yeast, Sld7/Sld3, Dbp11 and Sld2 regulate the maturation of the pre-RC into the functional helicase complex Cdc45-MCM-GINS with pol ε (CMGE) in different DDK-CDK-dependent steps (for review, see[10]). Their respective counterparts in metazoans, MTBP (Mdm2-binding protein)/Treslin, TopBP1 and RecQL4 are loaded onto the pre-RC to build the pre-initiation (pre-IC) complex[11–15]. The activation of the replicative helicase is the crucial step during origin firing. The competition between origins for limiting replication factors such as Sld3, Sld2, Dbp11, and Dbf4 contributes to setting their firing time in budding yeast[16]. Treslin, RecQL4, TopBP1, and the embryonic DDK regulatory subunit Drf1 levels become limiting for replication following the increase in the nuclear to cytosolic ratios in vitro and in vivo after 12 cell cycles during *Xenopus* development[17].

Rif1 (Rap1-interacting factor 1) is a key regulator of the replication timing program[18] and modulates late replication in yeast[19,20], mice[21], and human cell culture lines[22]. In the absence of Rif1, genome-wide studies (Repli-Seq) revealed substantial switches of late RT domains becoming early and early RT domains becoming late[21–23]. Rif1 binds protein phosphatase 1 (PP1) and recruits it to phospho-sites on the pre-RC complexes, acting both negatively on origin activation by counteracting the phosphorylation of the replicative helicase complex MCM2–7 by DDK[24–26] and positively on the licensing step[27]. However, single-molecule (SM) analyses by DNA fiber stretching techniques led to contradictory conclusions on more local effects, such as origin activation and fork speed, in mammalian cells after Rif1 depletion[21,27]. Rif1 mostly coats broad late regions (Rif1-associated domains, RADs) in multicellular organisms[23], consistent with reduced helicase activation in late heterochromatin due to locally high phosphatase activity in *Drosophila*[28]. It also binds G-quadruplex-like sequences[29] and is attached to the nuclear lamina[23]. Rif1 regulates replication firing time as a possible organizer of the chromatin architecture, and it has been proposed that Rif1 is a molecular hub to co-regulate RT and nuclear architecture[30]. After Rif1 KO in human mESC cells, RT changes lead to chromatin modifications and genome compartment alterations, suggesting that Rif1 contributes to maintaining a global epigenetic state[31]. But how these different roles of Rif1 can be integrated with its central role in the eukaryotic replication program regulation has not been explored so far. In *Xenopus* egg extracts, Rif1 depletion led to damage-resistant DNA synthesis[32] and increased bulk DNA synthesis during normal S phase[25]. Recently, we have shown that Rif1 depletion in early embryos accelerates cell divisions and substantially increases the number of active forks during the early S phase in *Xenopus* egg extracts measured on single DNA fibers by DNA combing[9]. However, we did not investigate how Rif1 regulates the coordinated firing of replication origins.

Stochasticity is commonly accepted to play an essential role in the replication process of all eukaryotic organisms, from yeasts to humans[33,34]. Early origins fire, on average with a higher probability than late ones. To picture the complex process of DNA replication, numerical models represent the replication as the sum of spatio-temporal dynamics of limiting initiation factors and initiation probabilities[35–40]. Using this representation, we modeled the dynamics of the replication process in the unchallenged, checkpoint-inhibited, or Plk1-depleted *Xenopus* in vitro system[41,42]. This led us to propose that in this system, as in other eukaryotes, the genome would be segmented into regions of low and high probabilities of origin firing. These regions would therefore be reminiscent of RT domains characterized in differentiated cells.

Our model's generality motivated us to use it as a framework to analyze Rif1's role in DNA replication. We performed an in-depth analysis of our SM-data set from DNA fibers with numerical simulations and experimentally tested our model predictions after Rif1 depletion in the *Xenopus* in vitro system. We describe the dynamics of the Rif1 protein during early embryonic cell divisions and the S phase and found that after Rif1 depletion, more replication clusters are activated on combed DNA fibers. In contrast, origin distances are only moderately reduced within clusters, and the apparent fork speed is increased. The analysis of the experimental SM-data by our previously established model suggests that Rif1 depletion accelerates the temporal replication program. The model also suggests that Rif1 depletion increases the number of limiting initiation factors and regions with a high probability of origin firing; experiments provided us with further support for these predictions. Therefore, the dynamic and integrated picture of DNA replication we propose is in adequation with the Rif1-dependent control of the replication program during rapid embryonic S phases.

## Results and discussion

**Rif1 dynamics during early development and the S phase in *Xenopus*.** Studies in *Drosophila* embryos suggested that Rif1 is required for the late replication of satellite sequences after the mid-blastula transition (MBT)[28,43], when a different replication timing program is established at the onset of zygotic transcription. We first monitored the changes in Rif1 abundance in whole *Xenopus* embryos before and after MBT, which occurs between cell cycles 12 and 13 in this organism[44]. We found that Rif1 protein levels were high from fertilization and during the early cell cycles, decreased after the mid-blastula transition and remained low until the late larval stages (Fig. 1a, b). This intriguing observation suggested that the Rif1 abundance could play an important role during these early embryonic cycles when cells highly proliferate. We followed Rif1 recruitment to chromatin in the efficient, well-characterized in vitro replication system, mimicking the first S phase after fertilization (Fig. 1c). Upon

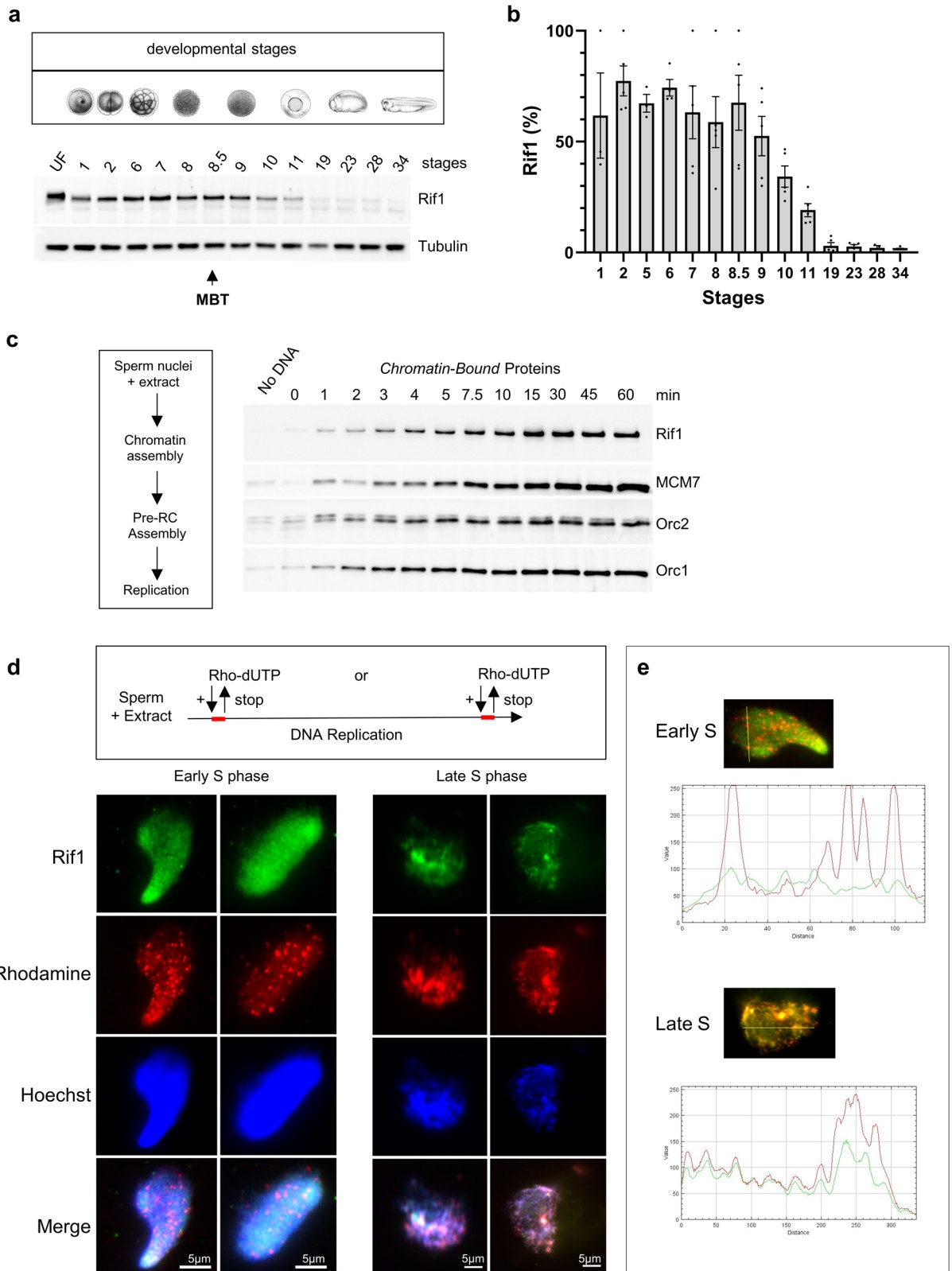

addition of sperm DNA to egg extracts, chromatin is assembled, replication proteins are imported and recruited on chromatin, and nuclei synchronously start DNA replication. We found that Rif1 accumulation to chromatin paralleled that of the ORC-complex, exhibiting a continuous increase until reaching a plateau; this observation further corroborates a role for Rif1 throughout the S phase in this model system. Next, we investigated the nuclear localization of Rif1 by immuno-fluorescence during early and late S phases in the in vitro system. In parallel, we pulse-labeled early and late replicating forks with rhodamine-dUTP to follow whether Rif1 could co-localize with ongoing replication (Fig. 1d, e). We found that Rif1 staining was not homogenous, co-localized with DNA, but not with early replication forks, as shown in mouse cells[21]. However, during the

**Fig. 1 Rif1 dynamics during early *Xenopus* development and in the S phase of the in vitro system. a** Time course analysis of Rif1 expression throughout development; whole embryo protein extracts were analyzed by western blotting against indicated proteins, tubulin was used as loading control; MBT (mid-blastula transition), UF (unfertilized egg). **b** Quantification of Rif1 abundance in three biological replicates and two technical replicates of western blot series of embryonic whole cell extract, plotted as mean OD normalized to Tubulin, scaled using min and max: xscaled = (x − xmin)/(xmax−xmin), Mean with SED, n = 5 (data points). **c** Time course analysis during the S phase; sperm nuclei were incubated in *Xenopus* egg extracts and chromatin was isolated for immunoblotting at indicated time points before and during DNA replication. **d** Rif1 nuclear localization during S phase; sperm nuclei were incubated in egg extracts, rhodamine-dUTP was added at the beginning of the incubation and stopped during the early S phase (30 min), left panel, or for 2 min at the end of S phase (60–62 min), right panel, reactions were stopped and nuclei were isolated and processed for immunofluorescence. **e** Fluorescence intensity profile plots along the indicated yellow line, to visualize colocalization between rhodamine and Rif1 for two example nuclei from (**d**).

late S phase, Rif1 staining at least partially overlaps with large replicating chromatin regions. This raises the question of whether Rif1 could participate in the temporal regulation of DNA replication in the *Xenopus* in vitro system, as shown in other organisms.

**Rif1 depletion increases initiation frequency and apparent fork speed during the early S phase.** To better understand how Rif1 could regulate the replication program in the in vitro system, we analyzed replication origin activation by DNA combing after immunodepletion of Rif1 from egg extracts (Fig. 2a). We previously focussed on comparing the role of Rif1 in reducing fork density[9]. Here, we enrich the DNA combing experiments with new data and a detailed, quantitative analysis in the absence of Rif1. We found that Rif1 depletion boosted mean DNA synthesis 4–8-fold in the early S phase (50–60 min) versus a two-fold increase mid-S phase (90–105 min) (Supplementary Fig. 1). This may result from increased initiation events, fork speed, or both. We calculated initiation frequencies (number of initiations per time unit per unit length of unreplicated DNA), $I(t)$, and found that initiations were about three-fold higher during the early S phase after Rif1 depletion compared to the control versus a two-fold increase at mid-S phase (Fig. 2b). Thus, Rif1 depletion led to a substantial increase in initiations, especially during the early S phase. We, therefore, expected to observe a sharp decrease of origin distances. However, we found that measured origin distances (eye-to-eye distances, ETEDs) remained unchanged during the early S phase and only moderately decreased in mid S phase (Fig. 2c, median decrease 1.1–1.4 fold) in the absence of Rif1. The apparent discrepancy between the global initiation frequency and ETEDs measured on single fibers can be explained by the limit in fiber length set by DNA breaks or microscope field and the organization of 2–8 origins per cluster[5,6]. To study differences in cluster activation in both conditions, we analyzed the distribution of all replication tracks per fiber as described[6]. Both distributions contained an excess of fibers with either no eye or multiple eyes compared to a random distribution (Supplementary Fig. 2a, b), consistent with the fact that origins are not activated independently of each other but in clusters. Moreover, the distributions from Mock and Rif1 depleted extracts differed significantly (Fig. 2d; *p* value = 4.77 $10^{-32}$, Mann–Whitney test). After Rif1 depletion, we observed that the percentage of fibers without replication tracks was reduced during the early S phase. Conversely, the percentage of fibers containing more than 2 labeled replication tracks per fiber sharply increased. Importantly, fibers showing more than 5 tracks were 5–10-fold more frequent, suggesting either larger clusters or more activated clusters in the absence of Rif1 (representative fibers in Supplementary Fig. 3a). Given that incomplete eyes or gaps are excluded from ETED measured on fibers (or intra-cluster ETED), it follows that large inter-cluster ETEDs have a higher probability of being excluded. We thus also analyzed these excluded ETED (or inter-cluster ETED) for both conditions (Supplementary Fig. 3b, Supplementary Table 1), using a method described[6]. We observed that these

distances were 5.3–1.2-fold larger from the very early to the mid S phase, respectively, in controls compared to Rif1-depleted extracts. This reflects that, depending on S phase progression, 5.3–1.2 more replication clusters were activated in the absence of Rif1. The observed changes in the replication track patterns, the inter-cluster distances (excluded distances), together with the moderate changes in the intra-cluster ETED, suggest that the initiation increase after Rif1 depletion was mainly caused by the activation of whole replication clusters. In mammalian cells, DNA fiber analysis led to somehow contradictory observations in the absence of Rif1. Whereas no change in origin distances was observed in mouse cells[21], an increase was detected in human cell lines, attributed to larger chromatin loop sizes[22]; another study described a decrease of origin distances due to the degradation of Orc1[27] in the absence of Rif1. These different observations across species suggest that the effect of Rif1 at the level of single origins appears minor compared to its strong global impact at the level of chromatin domains.

Next, we noticed that replication eye length (EL) distributions were significantly shifted towards larger sizes at the early S phase after Rif1 depletion compared to the control (Fig. 2e, 1.4–2-fold median increase), but not during mid-S phase, pointing to an increase in the apparent fork speed during early S phase. This increase could be either due to a substantial increase in fork speed or the consequence of new initiations close to active forks, which led to replication eye merging. Some studies in mammalian cells and *Drosophila* showed an increase in fork speed in the absence of Rif1, measured by increased eye length or copy number analysis[22,25,45]. However, other mammalian cells' DNA fiber studies revealed little or no changes in fork speed[21,25,27]. We conclude that Rif1 depletion strongly accelerates DNA synthesis in egg extracts by advancing the activation of replication clusters at the beginning of the S phase, whereas it leads to only a moderate increase in origin activation inside clusters and an increase in the apparent fork speed.

**Rif1 depletion globally accelerates DNA replication kinetics and increases early replication foci.** To provide a more comprehensive picture of the role of Rif1 in DNA replication, we fitted our recent minimal dynamic model of DNA replication, which had proven its robustness to simulate the replication process under different experimental conditions[41,42], to our Mock and Rif1 depletion combing data. This numerical model assumes that origin firing is stochastic and is driven by the initial number of the limiting replication factors $N_0$, interacting with potential origins and the import rate $J$ of these limiting replication factors (Fig. 3a). The genome needs to be divided into regions with a high probability of origin firing, $P_{in}$, in the fraction $\theta$ of the genome, and regions with a lower probability of firing, $P_{out}$, located in the remaining fraction 1-$\theta$. The fraction $\theta$ of the genome corresponds to regions with efficient and synchronous origins, mainly early firing. In contrast, fraction 1- $\theta$ corresponds to regions with inefficient, mostly dispersed, essentially late-firing origins. Finally, replication forks promote other origin firing with

**a**

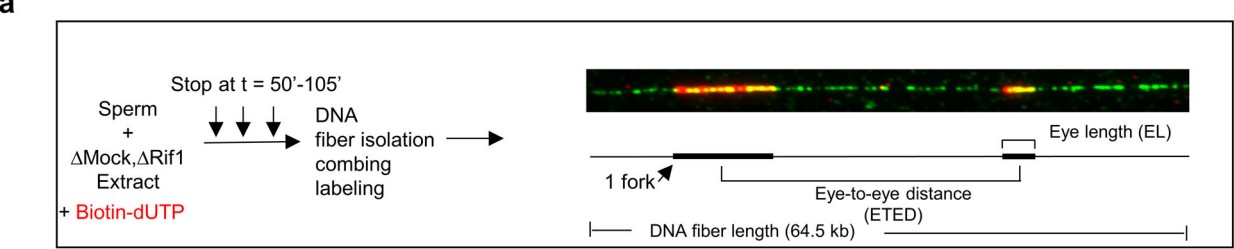

**b**

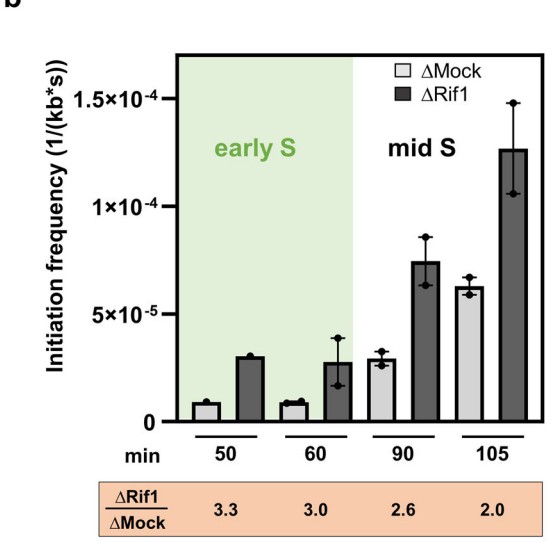

**c**

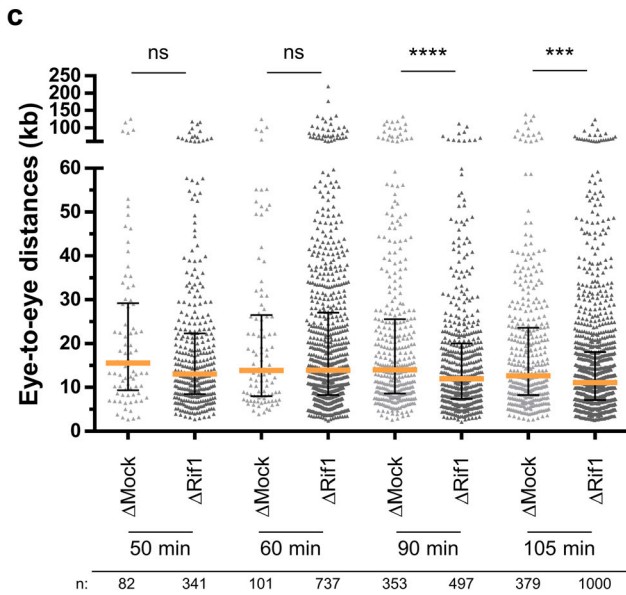

**d**

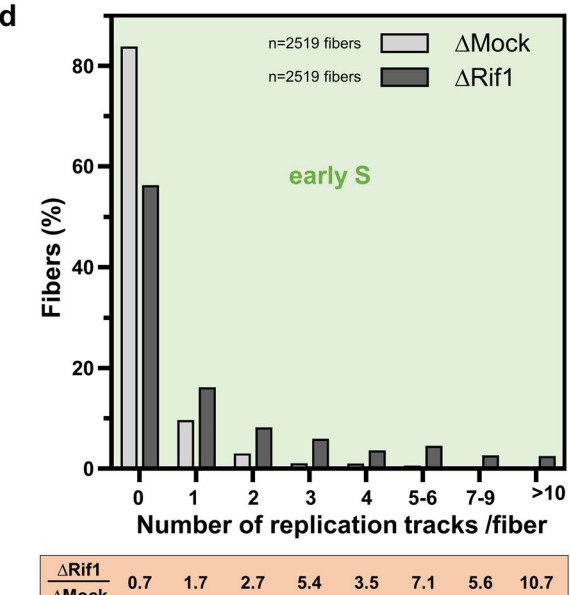

**e**

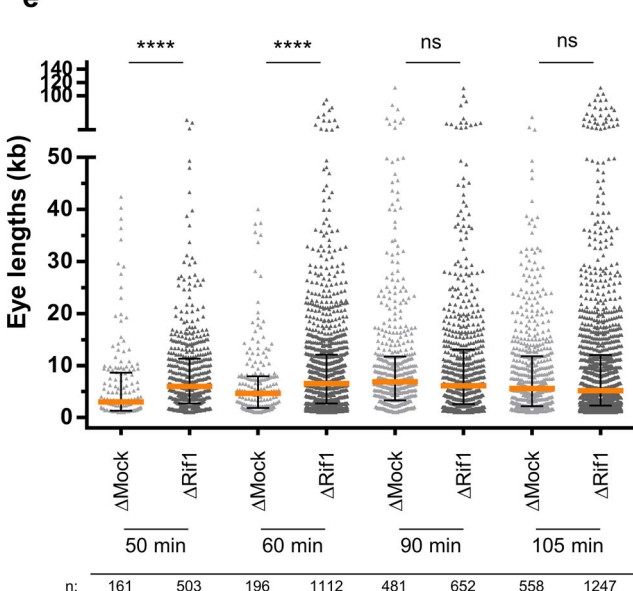

a probability $P_{loc}$ within a distance $d$ from the fork. For control and Rif1 depleted conditions, the model well reproduced the replication parameters (Supplementary Fig. 4a–c). We found that all replication parameters changed when Mock and Rif1 depletions were compared at the same time point (90 min), demonstrating the dramatic effect of Rif1 depletion seen by DNA combing experiments: on one hand, $\theta$, $P_{out}$, and to a lesser extent,

$P_{in}$ and $N_0$ significantly increased (Fig. 3b). On the other hand, $J$, $d$, and $P_{loc}$ decreased. Altogether, this suggests that Rif1 depletion may lead to (i) an increase in the fraction of regions with highly efficient, early origins ($\theta$), (ii) an increase in origin firing for late dispersed origins ($Pout$) and (iii) a small increase of origin firing inside early regions ($Pin$). We decided to test experimentally whether Rif1 depletion increases $\theta$, meaning advances replication

**Fig. 2 Rif1 depletion strongly increases origin activation in the early-S phase at the level of replication clusters. a** Principle of DNA combing experiment with a fiber example and measured parameters. Sperm is replicated in control (ΔMock) or Rif1 immunodepleted (ΔRif1) egg extract in the presence of biotin dUTP, DNA was isolated at the indicated times, and then DNA combing was performed in two independent experiments. **b** Mean initiation frequencies ($I(t)$) with standard error of mean with SEM, $n = 2$ (data points) and ratio ΔRif1/ΔMock ($I(t)$) were calculated. **c** Scatter dot plot of eye-to-eye distances (ETED) at different time points from replicate 1, median with interquartile range, Mann–Whitney test. **d** Percentage of unreplicated fibers and fibers with increasing number of replication tracks per fiber from early S phase from both independent experiments, with an equivalent fiber length distribution in mock and Rif1 depleted condition; percentage ratios indicated below each class. ($n = 2519$ fibers for each distribution, Mann–Whitney test on distribution, $p$ value $4 \times 10^{-41}$). **e** Scatter dot plot of eye lengths (EL), replicate 1, median with interquartile range, Mann–Whitney test; n indicated below, ns: non-significant, * indicates significant difference (Mann–Whitney U test, two-sided, $p < 0.05$: $p$ values: * 0.01–0.05; ** 0.001–0.01; *** 0.0001–0.001; **** <0.0001).

at the domain level. Replication foci probably correspond to the genomic replication domains in metazoans[46]. In the absence of Rif1, the replication foci pattern changed in mice and human cells[21,22]. Temporally defined, albeit more rudimentary, replication foci patterns can also be distinguished in the *Xenopus* in vitro system[7]. We, therefore, performed a quantitative analysis of replication foci after short pulse labeling with rhodamine-dUTP, feasible at the very early S phase when foci can be quantified in this experimental system. This analysis showed a significant, two-fold increase of the mean replication foci number after Rif1 depletion compared to the control (Fig. 3c, d, Supplementary Fig. 5). It suggests that in the absence of Rif1, more chromatin domains are activated early, in agreement with the effect in our numerical model.

However, we found in our previous work that $J$, $\theta$, $Pout$, and $d$ model parameters also vary in the same range and dynamics during normal S phase progression[41]. This suggests that Rif1 depletion would accelerate the S phase by compressing the replication program without changing many replication parameters per se. In other words, Rif1 depletion would lead to a more homogenous temporal program in the *Xenopus* system.

If this is true, local parameters should not differ after Rif1 depletion if they are compared at equivalent replication extents. We, therefore, pooled DNA fibers from the two replicates and all time points, classified them according to their replication extent (replicated fraction $f$) (Fig. 4a), and then compared different replication parameters in the presence and absence of Rif1 (Fig. 4b). No or only little significant differences in eye-to-eye distances, eye lengths and fork density were observed between Mock and Rif1 depleted conditions at the level of single fibers in this analysis. The initiation frequency increased slightly slower after Rif1 depletion, which could illustrate a more homogenous origin activation inside clusters. Altogether, these results strongly suggest that Rif1 depletion affects the replication program by a global acceleration at the level of chromatin domains but with only few changes of replication parameters locally.

**Rif1 depletion leads to an increase of DDK and origin firing factors on chromatin.** Next, we asked whether in silico parameters would change after Rif1 depletion when compared at an equivalent replication extent (Fig. 5a, Supplementary Fig. 4a, c). We found no significant increase for *Pin*, *Pout* and $\theta$, again consistent with a global acceleration after Rif1 depletion. However, the number of limiting initiation factors, *N0*, still increased in the absence of Rif1, suggesting that Rif1 lowers the number of initial initiation factors independent of S phase progression.

It was uncertain whether Rif1 could regulate limiting initiation factors, essential for maturing the pre-replication complexes into the pre-initiation complexes in vertebrates[11,14]. To experimentally test our unexpected in silico prediction, we analyzed by western blotting the effect of Rif1 depletion on the binding to chromatin of several origin firing factors (Treslin, MTBP, TopBP1, RecQL4) and the S phase kinase Cdc7/Drf1 (DDK).

After Rif1 depletion, the total amount of chromatin-bound Treslin, MTBP, Cdc7, Drf1, RecQL4, and Cdc45 significantly increased in three replicates, whereas TopBP1 slightly decreased throughout the S phase (Fig. 5b, c, Supplementary Fig. 6a). The increase of the pre-IC proteins Treslin/MTBP, RecQL4 and Cdc45 is consistent with the increased origin activation we observed after Rif1 depletion. The increased chromatin binding of Cdc7/Drf1 after Rif1 depletion could be an indirect effect related to Rif1's role in regulating nuclear architecture, as it was suggested in mammalian cells, where DDK accumulation was observed to a lesser extent[22] than we show here in *Xenopus*. Alternatively, it could also be due to a direct inhibition of DDK by Rif1/PP1; in budding yeast, an interaction between Rif1 and Dbf4 has been observed, but the removal of Rif1 did not alter DDK activity[24,47]. The moderate decrease of TopBP1, also observed in a former study[32], could be linked to its replication-independent roles[48]. In the absence of Rif1 and PP1, slower migrating forms of hyperphosphorylated MCM4 were detected, as demonstrated in a former study in *Xenopus*[25]. We have previously shown that Treslin and MTBP are phosphorylated in a DDK-dependent manner[42]. If Rif1 would recruit PP1 also to pre-initiation complexes, we expected that MTBP/Treslin would remain phosphorylated in the absence of Rif1/PP1. We found that the slower migrating forms of Treslin and MTBP were enriched on chromatin compared to the quicker migrating forms in the absence of Rif1 in Fig. 5b and in Fig. 5d, respectively. Dephosphorylation by λ-phosphatase treatment confirmed that the slower migrating bands after Rif1 depletion represents a phosphorylated form of MTBP or Treslin (Supplementary Fig. 6b). These results show that the phosphorylation state of Treslin and MTBP depends on the presence of Rif1 in *Xenopus*. A similar effect was described in budding yeast for the Treslin homolog Sld3[47]. This observation could be explained by Rif1 recruiting PP1 or another phosphatase on the pre-initiation complex. Alternatively, maybe not exclusive, this could be the consequence of a Rif1-dependent inhibition of DDK recruitment (or activity), since an enrichment of DDK was observed after Rif1 depletion. The significance of the DDK-dependent phosphorylation and the Rif1-dependent dephosphorylation on Treslin/MTBP activity remains to be explored in future studies; MTBP phosphorylation by CDK or checkpoint kinases regulates origin firing, possibly by controlling the stability of the MTBP-Treslin complex[12,42,49]. Altogether, our observation of proteins enriched on chromatin after Rif1 depletion provides strong support for the prediction of the numerical model that Rif1 limits initiation factors. We were not able to generate a recombinant form of Rif1 that could rescue the depleted extracts. However, vertebrate Rif1 is a very large protein and is difficult to produce in an active form in recombinant systems[50,51]. Immunodepletion of Rif1 does not noticeably decrease the abundance of TopBP1 and other known checkpoint regulators in egg extracts[32], and two different proteomics approaches in *Xenopus* and mouse have not revealed Rif1 interactions with known negative replication regulators other

**a**

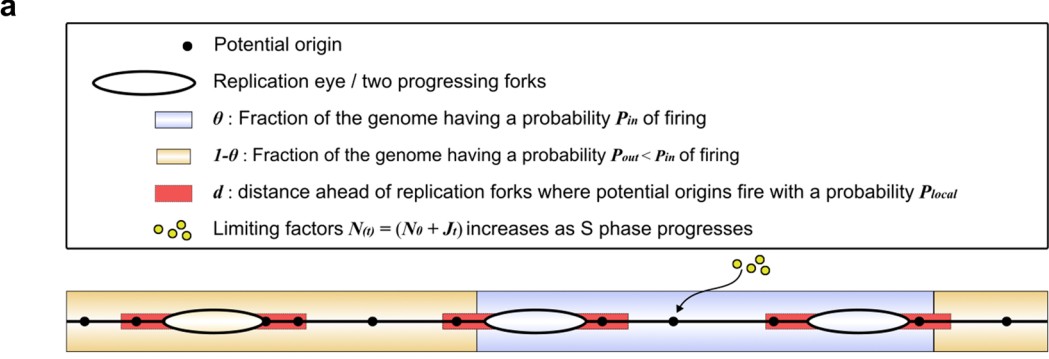

**b**

### Numerical simulation: Identical time (90 min)

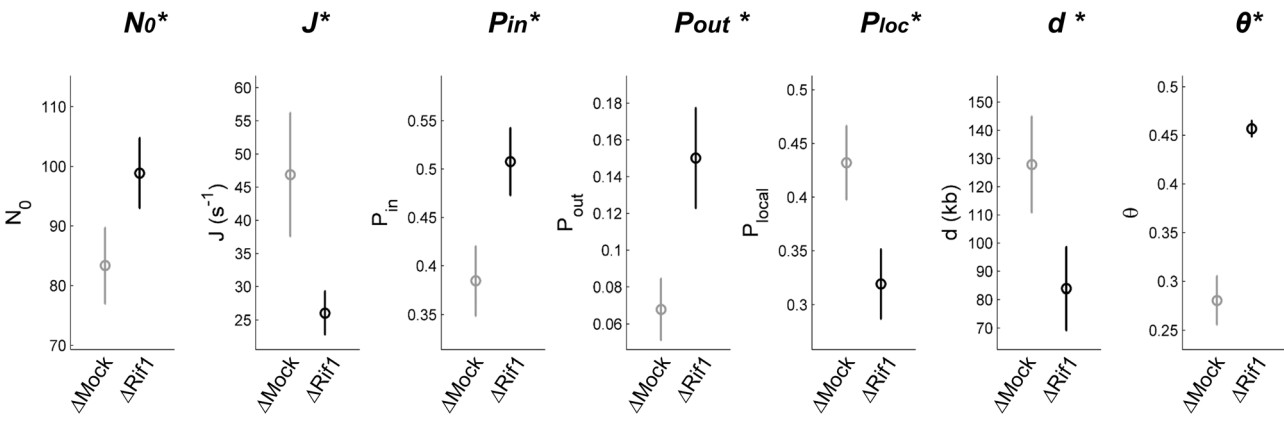

**c**

**d**

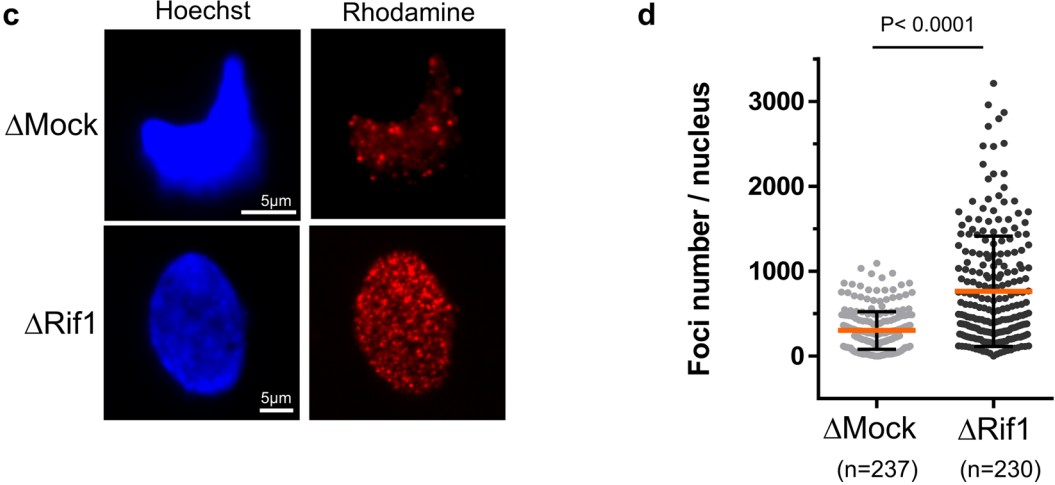

**Fig. 3 Rif1 depletion changes all in silico replication parameters in numerical simulations and increases the number of early replication foci. a** Diagram of the numerical model[41] with parameters described in the text. **b** Inferred model parameters by fitting ΔMock and ΔRif1 depleted combing data of two replicates for the same time interval (90 min). Circles indicate the mean value of the parameters over 100 different runs of the genetic algorithm; the error bars are standard deviations, (*) indicates a significant difference between the ΔMock and ΔRif1 samples, $\chi^2$ test. **c** Nuclei were incubated in Mock or Rif1 depleted extracts in the presence of rhodamine-dUTP and stopped early in S phase, representative images of nuclei by fluorescence microscopy. **d** Scatter dot plot of early replication foci number per nucleus, mean with SD, two-tailed Mann–Whitney test.

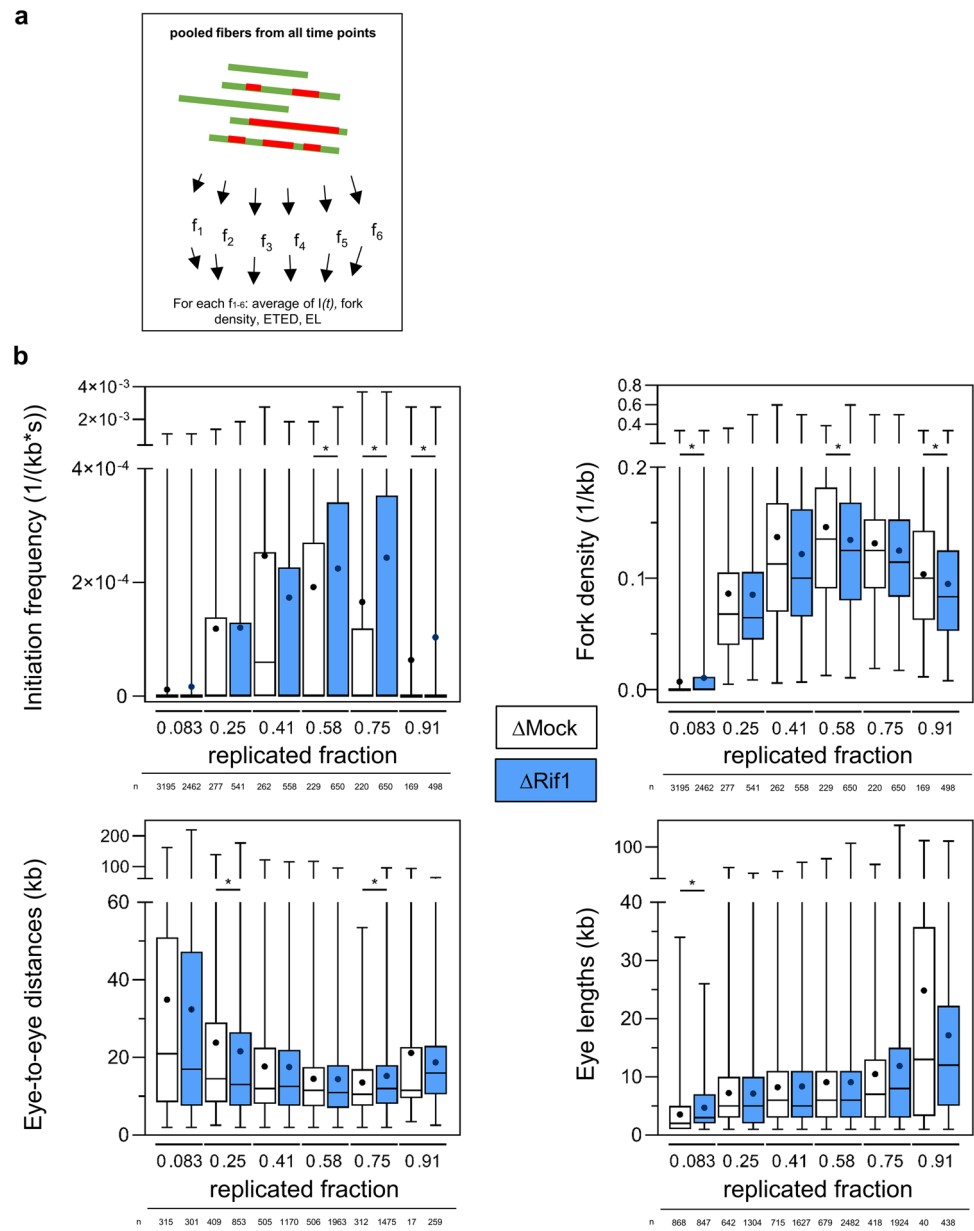

**Fig. 4 Rif1 depletion does not intrinsically change origin distances or eye lengths. a** Principle of analysis: All DNA fibers were grouped into six classes according to their replicated fraction $f_1$-$f_6$. **b** Box and whisker plots for initiation frequency ($I(f)$), fork density, ETED and EL after Mock and Rif1 depletion for each replicated fraction class, with medians (black lines) and means (black points), upper and lower quartiles, min and max, n indicated below, * indicates significant difference, Mann–Whitney U test, two-sided.

than PP1[25,52]. Finally, we analyzed MTBP protein levels in pre- and post-MBT embryos. We found that MTBP abundance is constant but declines after the MBT (Supplementary Fig. 7a, b), as shown in a proteomic study[53], when the S phase lengthens and the replication program slows down. This result suggests that MTBP could be, like its partner Treslin and other initiation factors[17], concentration limiting for the replication program during Xenopus development.

The results of our study have several important implications for the replication program and the role of Rif1 in early embryos. First, the early embryonic S phase is not running at its fastest biochemical rate and can be accelerated by Rif1 depletion in vitro and in vivo (this study and[9]). Second, origin firing factors such as MTBP/ Treslin, Drf1, and RecQL4 do not seem to be concentration limiting per se in Xenopus egg extracts since, in the absence of Rif1, more firing factors and Drf1 are recruited to chromatin, and many

more origins can be activated. Rather than being concentration limited, we propose that DDK and firing factors might be excluded from chromatin by Rif1-compacted chromatin domains, or their activity might be regulated at some origins or regions by Rif1-dependent PP1 targeting. Another possibility could be that self-organizing Rif1-PP1 hubs break down when DDK/CDK levels reach a threshold level leading to a more or less synchronized origin firing in later replicating sequences as proposed for satellite sequences in *Drosophila*[28]. It has been suggested that one conserved role of the RT program in eukaryotes is to adapt the optimal number of replication forks to the number of available initiation factors[6]. In the absence of Rif1 in yeast and mammalian cells, some early origins or regions fire late, consistent with the hypothesis of the exhaustion of initiation factors as an indirect result of unscheduled origin firing[20,21]. In egg extracts, the stockpile of maternal proteins may be sufficiently enriched in initiation factors

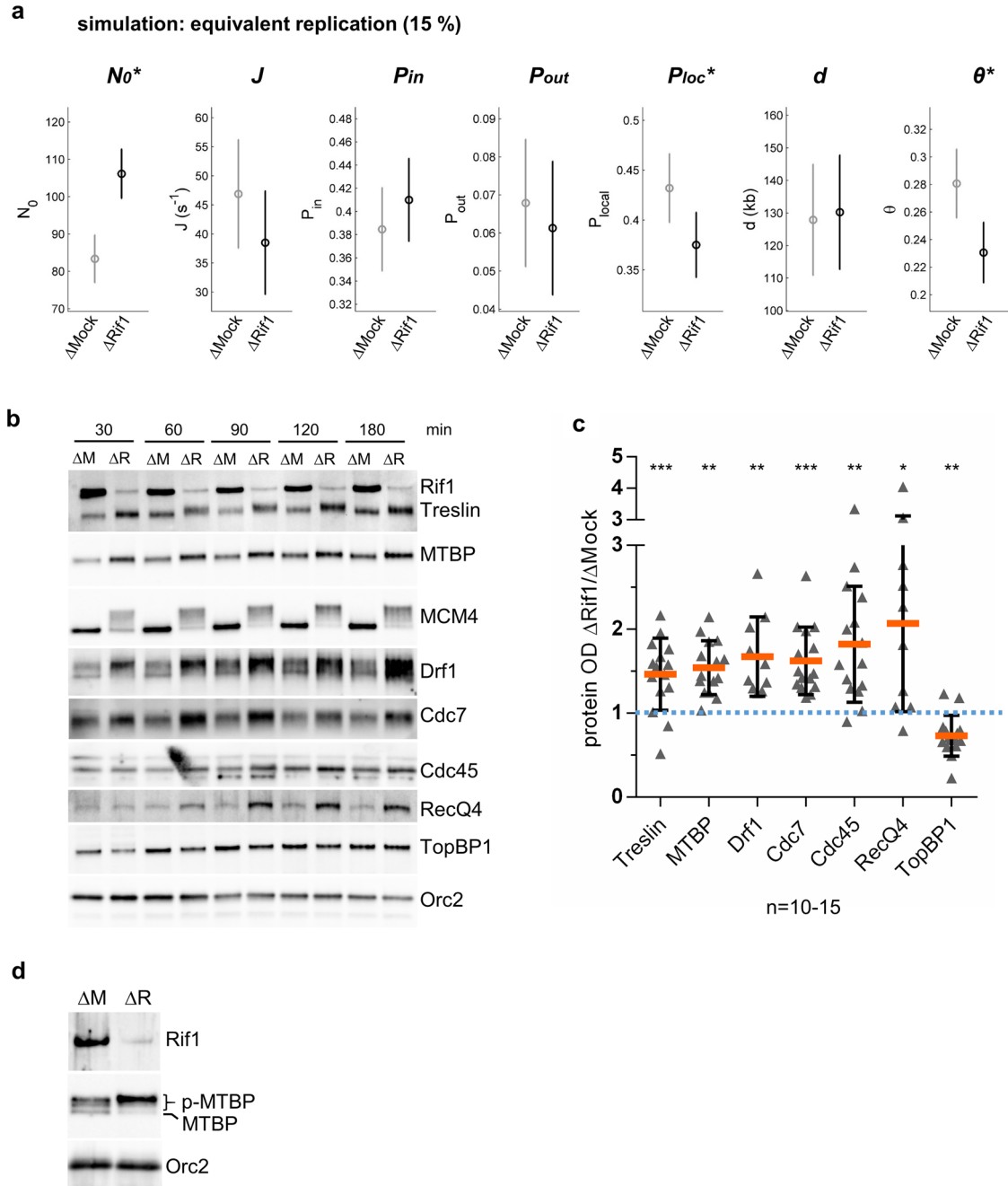

**Fig. 5 Rif1 depletion increases the chromatin recruitment of initiation factors. a** Inferred model parameters by fitting ΔMock and ΔRif1 depleted combing data of two replicates at a comparable percentage of replication. Circles indicate the mean value of the parameters over 100 different runs of the genetic algorithm; the error bars are standard deviations, (*) indicates a significant difference between the ΔMock and ΔRif1 samples, $\chi^2$ test as described[41]. **b** Chromatin fractions after Rif1 or Mock depletion were isolated at indicated times and analyzed by western blot for indicated proteins. **c** Ratios of ΔRif/ΔMock optical densities (OD) of western blot bands for indicated proteins from three independent chromatin experiments, normalized to Orc2-OD, scatter dot plot with mean and SD, including all time points ($n = 10$–15), one sample $t$-test. * indicates significant difference. **d** Phosphorylation of MTBP after Rif1 depletion, chromatin proteins were separated after 60 min using Anderson-SDS-PAGE to better resolve phosphorylated forms of MTBP[42].

to support unscheduled origin firing without slowing down origin activation. In support of this, we and others showed that replication factors become limiting for the replication program only at higher nuclei concentrations and at the MBT[17,54]. Last but not least, our results point to the role of Rif1 in regulating the replication timing program at the level of larger chromatin domains in early embryos because the major effect after Rif1 depletion is an increase in the number of (i) replication foci, (ii) early replication clusters on DNA fibers and (iii) early domains in the numerical model.

We propose a model for Rif1 function in early embryos where Rif1 binding to mid-late regions or domains restricts the accessibility of initiation factors to chromatin by keeping close domain contacts[23] and by establishing locally high phosphatase activity, which slows down S phase progression (Fig. 6). This could be by directly inhibiting initiation factors like Treslin/MTBP and/or indirectly by modulating DDK access to chromatin and the ability of MCM2-7 to compete for limiting factors. In the absence of Rif1, structurally more relaxed domains[23] and the

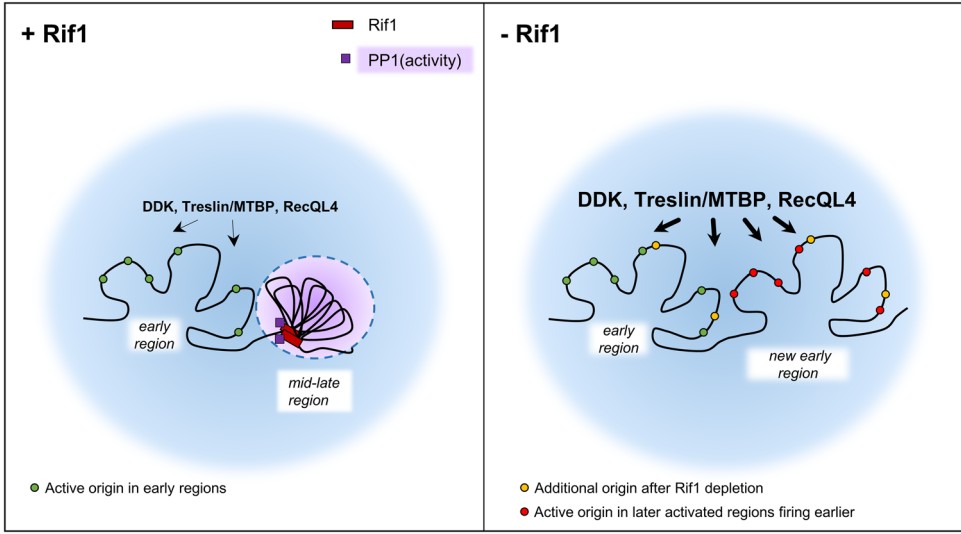

**Fig. 6 Model for Rif1 restraining the spatio-temporal replication program in early *Xenopus* embryos.** Rif1 orchestrates the temporal availability of initiation factors and PP1 in early *Xenopus* embryo's DNA replication at the level of domains; for details, see text.

absence of PP1 would allow the binding of DDK kinase and firing factors, such as Treslin/MTBP and RecQL4, to origins and would result in a boost of initiations in whole clusters, as we observed by DNA combing analysis.

## Conclusions

Our experimental approaches, supported by mathematical modeling, comprehensively describe a framework for the function of Rif1 in regulating origin activation. Using SM-analysis, we demonstrate that Rif1 depletion leads to a more homogenous temporal replication program in early *Xenopus* embryos through regulation at the level of replication clusters and chromatin domains and control of the replication origin firing factors Treslin/MTBP, DDK and RecQL4. Contrary to the prevailing opinion that replication kinetics is at its maximum in this experimental system, our study clearly shows that Rif1 slows down DNA replication and that an RT program exists. Since Rif1 is not essential in *Xenopus*[9], as in most other organisms, this leaves the question of the role of the RT program and Rif1 in early embryos. Early embryos may avoid activating too many origins in a given time window to minimize replication-associated DNA damage or topological stress, which would induce too much genomic instability in the absence of efficient DNA damage checkpoints in vivo. In addition, by regulating chromatin access to replication factors, when specific subnuclear compartments and eu/heterochromatin differentiation have not yet been established in early embryos, Rif1's role could be to maintain a global epigenetic state, preparing the ground for changes occurring later during differentiation and development.

## Methods

**Ethics statement.** All animal experiments have been carried out in accordance with the European Community Council Directive of 22 September 2010 (2010/63/EEC). All animal care and experimentation were conducted in accordance with institutional guidelines, under the institutional license C 91-471-102. The study protocols were approved by the institutional animal care committee CEEA #59 and received authorization from the Direction Départementale de la Protection des Populations under the reference APAFIS#998-2015062510022908v2 for *Xenopus* experiments.

**Embryo collection.** *Xenopus laevis* embryos were obtained by conventional methods of hormone-induced egg laying and in vitro fertilization[55], staged according to[56], and raised at 22 °C.

**Antibodies.** A detailed list of the antibodies used in this study for immunohistochemistry (IHC), immunodepletion and western blot (WB) is provided in Supplementary Methods 1. Rabbit polyclonal XRif1 antibody against the C-terminal domain of XRif1 has been obtained as described[9].

**Replication of sperm nuclei in *Xenopus* egg extracts.** Replication-competent extracts from unfertilized *Xenopus* eggs and sperm nuclei from testes of male frogs were prepared as described[57]. Sperm nuclei (2000 nuclei/μl) were incubated in untreated, Mock or Rif1 depleted extracts in the presence of cycloheximide to inhibit translation (250 μg/ml, Sigma), energy mix (7.5 mM creatine phosphate, 1 mM ATP, 0.1 mM EGTA, pH 7.7, 1 mM MgCl$_2$). Reactions were stopped at indicated times and treated for chromatin purification or DNA combing.

**Rif1-Immunodepletion.** Rabbit anti-Rif1 antibody for Rif1 depletion (Covalab, home-made antibody), pre-immune serum for Mock depletion were coupled overnight at 4 °C to protein A Sepharose beads (GE Healthcare) in spin columns (#69705, Pierce). Coupled beads were washed three times in EB buffer (50 mM Hepes, pH 7.5, 50 mM KCl, 5 mM MgCl$_2$). Coupled beads were incubated for 30 min at 4 °C in egg extracts (ratio 1:1), and after one round, the extracts were re-incubated for another 30 min at 4 °C with coupled beads and washed four times in EB buffer.

**Western blot.** Western blots were conducted using standard procedures on *Xenopus* embryo protein extracts[58] using 3–8% NuPage Tris–Acetate gels (Invitrogen). For the analysis of chromatin-bound proteins, we used a protocol slightly modified from[59]. Briefly, reactions were diluted into a 13-fold volume of ELB buffer (10 mM Hepes pH 7.5, 50 mM KCl, 2.5 mM MgCl$_2$) containing 1 mM DTT, 0.2% Triton X100, protease inhibitors and phosphatase inhibitors. Chromatin was recovered through a 500 mM sucrose cushion in ELB buffer at 6780 g for 50 s at 4 °C, washed twice with 200 μl of 250 mM sucrose in ELB buffer, and resuspended in 20 μl SDS sample buffer. For in vitro dephosphorylation, chromatin samples were incubated with λ-phosphatase (NEB) for 30 min at 30 °C in the presence of protease inhibitors, according to the supplier's manual. Proteins were separated on 3–8% NuPage Tris–Acetate gels (Invitrogen) or on 8.5% SDS-PAGE prepared with a 120:1 acrylamide: bisacrylamide ratio to increase the band separation for MTBP. Proteins were transferred to 0.45 μm Immobilon®-P membranes, subsequently incubated with the indicated primary antibodies followed by the appropriate horseradish peroxidase-labeled antibodies (1/10000, Sigma-Aldrich or GE Healthcare, see Supplementary Methods 1). Immunodetection was performed using Super Signal West Pico or Femto Chemiluminescence Kit (Pierce) on a ChemiDoc Touch Imaging System (BioRad). Quantification was done using BioRad ImageLab software.

**Molecular combing of DNA and detection by fluorescent antibodies.** Biotin-labeled sperm DNA was extracted and combed as described[60] after Mock or Rif1 depletion. Biotin was detected with AlexaFluor594 conjugated streptavidin followed by anti-avidin biotinylated antibodies. This was repeated twice, then followed by mouse ssDNA antibody, AlexaFluor488 rabbit anti-mouse, and AlexaFluor488 goat anti-rabbit for enhancement[61]. Images of the combed DNA molecules were acquired and measured as described[60]. The fields of view were chosen at random. 4000–6000 DNA fibers (around 400 Mb) were analyzed for each

experiment (Supplementary Table 1). Measurements on each molecule were made using Fiji software[62] and compiled using macros in Microsoft Excel. Replication eyes were defined as the incorporation tracks of biotin–dUTP. Replication eyes were considered to be the products of two replication forks, and incorporation tracks at the extremities of DNA fibers were considered to be the products of one replication fork. Tracts of biotin-labeled DNA needed to be at least 1 kb to be considered significant and scored as eyes. When the label was discontinuous, the tract of unlabeled DNA needed to be at least 1 kb to be considered a real gap. The replicated fraction of each fiber was calculated as the sum of eye lengths (red tracks) divided by the total DNA length (green track). Fork density was calculated as the total DNA divided by the total number of forks. The midpoints of replication eyes were defined as the origins of replication. Eye-to-eye distances (ETED), also known as inter-origin distances, were measured between the midpoints of adjacent replication eyes. Incorporation tracks at the extremities of DNA fibers were not regarded as replication eyes but were included in the determination of the replication extent or replicated fraction, calculated as the sum of all eye lengths (EL) divided by total DNA. ETED and EL scatter plots were obtained using GraphPad version 6.0 (La Jolla, CA, USA). The frequency of initiation was obtained as the number of new initiations defined as replication eyes smaller than 3 kb divided by the length of unreplicated DNA and by the time interval in which a detectable initiation event can occur, that is $\Delta t = 180$ s considering a replication fork speed of ~0.5 kb/min[6] (see also Supplementary methods 2 for summary definitions of replication parameters).

**Numerical simulations.** A Monte Carlo method was used to simulate the DNA replication program in Matlab (versionR2020b); the results were compared to the DNA combing data for Mock- and Rif1-depleted samples by using a genetic optimization algorithm[41,42]. Briefly, in the simulation, the genome is constituted as a vector of $10^6$ blocks of value 0 for the unreplicated blocks and 1 for the replicated blocks. Each block represents 1 kb. Potential origins are randomly distributed along the genome with an average density of one origin per 2.3 kb[63]. Initiation is regulated by the encounter between a not activated origin and a limiting factor, whose initial number $N_0$ linearly increases with a rate $J$ during S phase progression. The probability of successful origin firing after an encounter between the limiting factor and a potential origin is heterogeneous over the genome: it corresponds to $P_{in}$ in a fraction $\theta$ of the genome and $P_{out}$ for the remaining $1 - \theta$ fraction. Moreover, we consider that the probability within a distance $d$ from an active replication fork is increased by a local initiation probability $P_{loc}$. Once an origin is activated, the two replication forks elongate in opposite directions of one block for each calculation step. The limiting factor is sequestered by the forks and is made available for further origin activation once two converging forks merge.

**Immunofluorescence and replication foci number analysis.** Sperm nuclei (2000/µl) were added to replication reactions in the presence of 20 µM rhodamine-dUTP (Roche) and stopped at indicated time points or pulse labeled for the stated time period, 20 µl aliquots were diluted in 500 µl PBS and fixed by the addition of 500 µl p-formaldehyde 8%. Nuclei were spun onto coverslips through 1 ml of 20% sucrose cushion in PBS. Coverslips were blocked in 1xPBS, 0.2% Triton X100 containing 20% goat serum for 3 h, washed and incubated overnight in the same buffer in the presence of anti-Rif1 serum (1/600)[9]. After washing, coverslips were incubated for 2 h at 37 °C in goat anti-rabbit Alexa Fluor 488 (A488) secondary antibody (1/200) in the presence of Hoechst 33258. After washing, mounted coverslips were imaged for Hoechst, rhodamine, and A488 using an Olympus BX63 fluorescence microscope. The Red and Green fluorescence intensity profiles were generated using Fiji software. Quantitative analysis of replication foci was adapted from[64].

**Statistics and reproducibility.** Statistical analysis (GraphPad Prism software, version 8.3.0 and Matlab, version R2020b) were performed using statistical tests with $p$ values and sample size as mentioned in the figures, legends and in Supplementary Data. All replicates indicated in legends are biological replicates. A $p$ value ≤ 0.05 was considered significant.

**Reporting summary.** Further information on research design is available in the Nature Portfolio Reporting Summary linked to this article.

## Data availability

All data are included in this paper and its supplementary information files. Numerical source data for all graphs and charts are provided in the Supplementary Data file. Uncropped Western blot images are presented in Supplementary Fig. 8. Other information about this study are available from the corresponding author upon reasonable request.

## Code availability

The code of simulation and multivariable fitting used in this study, initially published in[41], is deposited on GitHub (https://github.com/DiDiCi/MMsimulation).

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

## Acknowledgements

We thank J. Walter for the gift of XCdc7 antibody. Anti-*Xenopus* Orc2 antibody, developed by JL. Maller was obtained from the European *Xenopus* Resource Centre, curated with funding from the Wellcome Trust/BBSRC and maintained by the University of Portsmouth, School of Biological Sciences. We are grateful to A. Donval for her help with frog fertilizations. We thank B. Miroux and J.-L. Ferrat for critical reading of the paper. This research was supported by grants to OB from ARC (Association pour la Recherche sur le Cancer), Association Retina France, Fondation Valentin Haüy, and UNADEV (Union Nationale des Aveugles et Déficients Visuels) in partnership with ITMO NNP (Institut Thématique Multi-Organisme Neurosciences, sciences cognitives, neurologie, psychiatrie)/AVIESAN (alliance nationale pour les sciences de la vie et de la santé). D.C. had an IDEX Paris-Saclay University Ph.D. fellowship. Work in the laboratory of W.G.D. is supported by National Institutes of Health grant R01 GM043974.

## Author contributions

O.H. designed, performed, supervised and analyzed experiments and contributed to the writing of the paper, D.C. analyzed experiments and designed the model and the computational framework, H.N. performed experiments, O.B. designed and performed experiments, A.K. and W.G.D. provided critical reagents, A.G. designed the model and the computational framework and supervised the analysis, K.M. designed, performed, analyzed, supervised experiments and wrote the paper. All authors discussed the results and contributed to the final paper.

## Competing interests

The authors declare no competing interests.
