## [Peer Review File · Communications Biology]

Reviewers' comments:

Reviewer #1 (Remarks to the Author):

Review of COMMSBIO-23-0399 (Marheineke)

In this work, the authors have provided evidence that depletion of Rif1 from *Xenopus* egg extracts increases levels of DNA synthesis largely by promoting the activation of new replication clusters. They propose that this effect involves increased phosphorylation of the origin firing factor MTBP as well as elevated recruitment of this factor to chromatin. Overall, this study is informative and well-executed. It is an extension of work recently published by this group. I would recommend publication pending some minor revisions.

1. Treslin/Ticrr forms a complex with MTBP and is also regulated by phosphorylation. Did the authors examine the phosphorylation of Treslin/Ticrr following the depletion of Rif1? I think that data on this issue would strengthen the paper.

2. Although phosphorylation of MTBP increases following the depletion of Rif1, this observation does not necessarily mean that Rif1/PP1 acts directly on MTBP. The findings should be presented more cautiously.

Reviewer #2 (Remarks to the Author):

Haccard et al. investigate the role of Rif1 in the regulation of replication timing in the early embryonic cell cycles of *Xenopus laevis*. Contrary to the general assumption that these cycles proceed as fast as possible, the authors demonstrate that replication rates are increased in the absence of Rif1. Further, they show that Rif1 decreases the phosphorylation and chromatin binding of MTBP, a replication initiation factor, leading to the plausible hypotheses that Rif1-dependent dephosphorylation of MTBP reduces the rate of origin firing in Rif1-bound regions of the early embryonic genome. The work is well done, the paper well written, and the conclusion will be of interest to those working in the fields of nuclear cell biology and genome metabolism.

Although the authors results and conclusions are straight forward and compelling, they overstate the power of their mathematical model over emphasize the model that Rif1 regulates the accessibility of late-replicating chromatin, a model for which they provide no supporting evidence. Modifying their claims in these two area would lead to a more balanced and robust paper.

Their mathematical model of replication kinetics is a useful tool for developing and testing hypotheses, for instance to test if intuitive models are quantitatively plausible, or to suggest the response of parameters to experimental perturbation that could be experimental investigated. However, it is not able to robustly predict which parameters change response in a given experimental condition. It is said that with 4 free parameters, one can fit an elephant <https://en.wikipedia.org/wiki/Von_Neumann's_elephant>. The authors' model has 7 free parameters. In any case, they are generally appropriately cautious about conclusions drawn from their model. However, they overstate their case, on Page 11, Line 1, where they claim, about their model analysis, that "this demonstrates that Rif1 depletion mainly leads to ...". They should revise the phrase to " this analysis suggests that Rif1 depletion may lead to ...".

Speaking of models, one fits models to data, not vice versa. On Page 9, Line 15 they should replace "data to our" to "data with our".

The other subject on which the authors should be more measured is the hypothesis that Rif1 excludes limiting replication factors from late replicating chromatin. They provide no evidence that address how Rif1 regulates the function of limiting replication factors, but a simpler model is that it creates a zone of locally high phosphatase activity in regions of late replicating chromatin. On Page 15, Lines 8-9, they present both hypotheses, which is fine. However, in the Abstract, they only

present the exclusion hypothesis. They should rephrase the last sentence of the Abstract from "probably by excluding limiting replication factors, such as MTBP, from late replicating chromatin" to "possibly by inactivating limiting replication factors, such as MTBP, in, or excluding them from, regions of late replicating chromatin".

Reviewer #3 (Remarks to the Author):

General comments

The major claim of this paper is that Rif1 depletion strongly accelerates DNA synthesis in egg extracts by advancing the activation of replication clusters at the beginning of the S phase, whereas it only modestly increases origin activation inside clusters and increases the apparent fork speed. Authors also show that the phosphorylation of MTBP1 is accelerated in Rif1 depleted extracts.

Since the discovery of Rif1 as a crucial regulator of replication timing, a number of reports have examined the effects of Rif1 on replication timing profiles. It has been well established that the Rif1 affects the profile of replication timing domains; in most cases, midS-late replication domains are converted to early replicating domains, indicating replication timing changes in a domain-specific manners (corresponding to the "clusters of the origins discussed in this report). Therefore, it may not be surprising that the replication timing changes in "clusters". However, The precise mode of origin regulation by Rif1 in early embryos have not been known.

Authors presented data showing that Rif1 increases the "early clusters" but not the firing frequency of individual origins and does not affect the fork rate.

Combing analyses have been professionally conducted and the data of combing and its analyses are convincing. The data provide conclusive information that in the *Xenopus* extracts, firing efficiency and fork rate is increased, contrary to the expectation from some of the data from yeasts and mammals.

The data are important in that it illustrates the role of Rif1 in regulation of DNA replication in early embryos.

Suggestions for improvement

Authors presented data showing that Rif1 increases the "early clusters" but not the firing frequency of individual origins and does not affect the fork rate.

The data of combing and its analyses are convincing but there is no information regarding the clusters. How are they distributed along the genome, and how are they affected by Rif1 depletion (How many clusters on the genome and what portions of the "late-firing clusters" are converted to early firing clusters? What are the features of the affected clusters (Rif1 binding, epigenetic feature, transcription activities and so on). Some of these questions need to be addressed.

In authors previous publications, it was stated that the each *Xenopus* cluster contain 5- 8 origins and span 50-100 kb. The current DNA fibers would contain one or two clusters. If longer DNA fibers containing many more clusters can be visualized and compared between the mock and Rif1-depleted extracts, it could be more visually convincing and would provide additional information on the nature of origin clusters and its regulation by Rif1.

Concern on the conclusion

Authors conclude that Rif1 binding to mid-late regions or domains restricts the accessibility of initiation factors to chromatin by keeping close domain contacts and establishing locally high phosphatase activity, which slows down S phase progression. The model is related to what was proposed before and can explain the data presented. However, in the absence of the knowledge on the nature of the origin clusters and Rif1 binding profile in the egg extracts, Rif1 could be simply inhibiting the PP1 that counteracts the phosphorylation on preRC factors mediated by DDK, and thus regulating the association of initiation factors including MTBP1.

Other concerns

The MTBP data in Figure 5 appears to be preliminary. There is no information on the

phosphorylation induced by Rif1 depletion, or on how the time course of chromatin association of MTBP (in the absence of Rif1) is related to that of other factors (DDK, RecQL4, MCM10, Cdc45, TopBP1 and others).

There is no add-back experiment of Rif1. Authors cannot completely rule out the possibility that immunodepletion depleted unknown Rif1 associated proteins, and would be more reassuring if one data for the recovery experiment could be presented.

Minor points

Page 15, line 8; Rif -> Rif1

Response to the Reviewers

Rif1 restrains the rate of replication origin firing in *Xenopus laevis*

Haccard *et al.*

We thank the three reviewers for their careful reading of our manuscript and their constructive comments. Please find below our point-by-point answers (in blue), followed by the corresponding edited text in quotes if appropriate.

Modified Figures 2, 5, 6 and new Supplementary Figures 2, 3 and 6 are shown at the end of this document.

Fig.1	Unchanged
Fig.2	Modified
Fig.3	Unchanged
Fig.4	Modified
Fig.5	Modified
Fig.6	Modified
Sup. Fig. 1	Unchanged
Sup. Fig. 2	New
Sup. Fig. 3	New
Sup. Fig. 4	Old Sup. Fig. 2
Sup. Fig. 5	Old Sup. Fig. 3
Sup. Fig. 6	New, Replaces old Sup. Fig. 4
Sup. Fig. 7	Old Sup. Fig. 5

Reviewers' comments:

Reviewer #1 (Remarks to the Author):

Review of COMMSBIO-23-0399 (Marheineke)

In this work, the authors have provided evidence that depletion of Rif1 from *Xenopus* egg extracts increases levels of DNA synthesis largely by promoting the activation of new replication clusters. They propose that this effect involves increased phosphorylation of the origin firing factor MTBP as well as elevated recruitment of this factor to chromatin. Overall, this study is informative and well-executed. It is an extension of work recently published by this group. I would recommend publication pending some minor revisions.

1. Treslin/Ticrr forms a complex with MTBP and is also regulated by phosphorylation. Did the authors examine the phosphorylation of Treslin/Ticrr following the depletion of Rif1? I think that data on this issue would strengthen the paper.

Answer: We thank the reviewer for this suggestion. Using a specific *Xenopus*-Treslin antibody, we found that the migration of Treslin in SDS-PAGE was retarded in the absence of Rif1 compared to the control, which now strengthens our initial findings on MTBP since they form a complex. Please note that the large MW of Treslin (220 kDa) makes it difficult to visualize its phosphorylations; we used NuPAGE 3-8 % gels and reproducibly detect these changes. We previously showed that Treslin and MTBP are phosphorylated in a Cdc7-dependent manner (Ciardo et al. 2021,NAR). Therefore, this observation suggests that Rif1/PP1, directly or indirectly, counteracts the Cdc7-dependent phosphorylations of the Treslin/MTBP complex. Consistently, observations showed that the yeast homolog of Treslin, Sld3, is phosphorylated in a DDK-dependent manner and dephosphorylated in Rif1 deficient cells (Mattarocci et al., 2014). In addition to the increase of Treslin phosphorylation, we found an increase in its recruitment on chromatin similar to that of MTBP. We added the Treslin blot in new

Fig. 5b and its quantification in new Fig. 5c, along with blots for other proteins (Cdc7, Drf1, RecQL4, Cdc45, TopBP1) as requested by referee 3 (see also new Supplementary Fig. 6a for the second replicate). We also added the dephosphorylation of chromatin-bound Treslin by phosphatase treatment in new Supplementary Fig. 6b, along with the dephosphorylation of MTBP for the same experiment. After the dephosphorylation by Lambda-Phosphatase, we observed a faster migration of Treslin, reaching the same level in mock and Rif1-depleted extracts. This shows that Treslin is phosphorylated and can be entirely dephosphorylated by phosphatase treatment, consistent with many published studies. Please note that on this blot, the Treslin shift between mock and Rif1 depletion, visible in the new Fig. 5b, is not apparent in the new Supplementary Fig. 6b because we used a different SDS PAGE (8.5% SDS-PAGE, see Methods section), which was aimed to resolve better the phosphorylation of MTBP (size 120 kDa).

We modified the text on page 8, line 31-page 9, line 13:

"In the absence of Rif1 and PP1, slower migrating forms of hyperphosphorylated MCM4 were detected, as demonstrated in a former study in *Xenopus*²⁵. We have previously shown that Treslin and MTBP are phosphorylated in a DDK-dependent manner⁴². If Rif1 would recruit PP1 also to pre-initiation complexes, we expected that MTBP/Treslin would remain phosphorylated in the absence of Rif1/PP1. We found that the slower migrating forms of Treslin and MTBP were enriched on chromatin compared to the quicker migrating forms in the absence of Rif1 in Fig. 5b and in Fig. 5d, respectively. Dephosphorylation by λ -phosphatase treatment confirmed that the slower migrating bands after Rif1 depletion represents a phosphorylated form of MTBP or Treslin (Supplementary Fig. 6b). These results show that the phosphorylation state of Treslin and MTBP depends on the presence of Rif1 in *Xenopus*. A similar effect was described in budding yeast for the Treslin homolog Sld3⁴⁷. This observation could be explained by Rif1 recruiting PP1 or another phosphatase on the pre-initiation complex. Alternatively, maybe not exclusive, this could be the consequence of a Rif1-dependent inhibition of DDK recruitment (or activity), since an enrichment of DDK was observed after Rif1 depletion. The significance of the DDK-dependent phosphorylation and the Rif1-dependent dephosphorylation on Treslin/MTBP activity remains to be explored in future studies; MTBP phosphorylation by CDK or checkpoint kinases regulates origin firing, possibly by controlling the stability of the MTBP-Treslin complex^{12,42,49}. Altogether, our observation of proteins enriched on chromatin after Rif1 depletion provides strong support for the prediction of the numerical model that Rif1 limits initiation factors. "

2. Although phosphorylation of MTBP increases following the depletion of Rif1, this observation does not necessarily mean that Rif1/PP1 acts directly on MTBP. The findings should be presented more cautiously.

Answer: We agree with the referee that the effect of the Rif1 depletion, thus the absence of chromatin-bound PP1 could be indirect on MTBP. Other phosphatases or kinases might be implicated in regulating the Treslin/MTBP complex. Our new data (Fig. 5b, c) show that DDK is also enriched upon Rif1 removal. The increased phosphorylation of MTBP and Treslin on chromatin could, therefore, also result from increased DDK on chromatin.

We now discuss this point in the text on page 9, line 5-8:

"This observation could be explained by Rif1 recruiting PP1 or another phosphatase on the pre-initiation complex. Alternatively, maybe not exclusive, this could be the consequence of a Rif1-dependent inhibition of DDK recruitment (or activity), since an enrichment of DDK was observed after Rif1 depletion."

Reviewer #2 (Remarks to the Author):

Haccard et al. investigate the role of Rif1 in the regulation of replication timing in the early embryonic cell cycles of *Xenopus laevis*. Contrary to the general assumption that these cycles proceed as fast as possible, the authors demonstrate that replication rates are increased in the absence of Rif1. Further,

they show that Rif1 decreases the phosphorylation and chromatin binding of MTBP, a replication initiation factor, leading to the plausible hypotheses that Rif1-dependent dephosphorylation of MTBP reduces the rate of origin firing in Rif1-bound regions of the early embryonic genome. The work is well done, the paper well written, and the conclusion will be of interest to those working in the fields of nuclear cell biology and genome metabolism.

1. Although the authors results and conclusions are straight forward and compelling, they overstate the power of their mathematical model over emphasize the model that Rif1 regulates the accessibility of late-replicating chromatin, a model for which they provide no supporting evidence. Modifying their claims in these two area would lead to a more balanced and robust paper.

Answer: We agree with the referee that a numerical model does not replace but only supports experiments. However, a numerical model that is process-based, as in our case, can infer some behaviors and guide the experimental procedure to verify or reject their reality. This strategy allows to validate or disqualify a processed-based model by experimental observations. We believe that our experimental verifications support most of the model inferences. We might have overstated the power of the model in some places. We made the change requested in point 2, on page 11, as well as in several other sentences in the revised version of the manuscript (see the modified version with track change).

In addition, our new results in Fig. 5b, c now strengthen that Rif1 depletion increases the availability of DDK (Cdc7 and Drf1), Treslin, RecQL4 and Cdc45, on top of MTBP. We now show this on new western blots with their quantifications obtained from three independent experiments. The increase of MTBP/Treslin and RecQL4, and the strong increase of DDK on chromatin after Rif1 depletion have never been described before in *Xenopus* or other systems and are, therefore very intriguing. Dbf4/Drf1, Treslin, and RecQL4 are limiting replication factors described in yeast and *Xenopus* (Mantiero et al., 2011, Collart et al., 2013). Altogether, these new results support the predictions of the numerical model ascribing a role for Rif1 in limiting initiation factors, either by direct inhibition or indirectly by excluding them through the known structural role of Rif1 in the nucleus.

We modified the text accordingly on page 8, line 18 to page 9, line 13.

“To experimentally test our unexpected *in silico* prediction, we analyzed by western blotting the effect of Rif1 depletion on the binding to chromatin of several origin firing factors (Treslin, MTBP, TopBP1, RecQL4) and the S phase kinase Cdc7/Drf1 (DDK). After Rif1 depletion, the total amount of chromatin-bound Treslin, MTBP, Cdc7, Drf1, RecQL4, and Cdc45 significantly increased in three replicates, whereas TopBP1 slightly decreased throughout the S phase (Fig. 5b-c, Supplementary Fig. 6a). The increase of the pre-IC proteins Treslin/MTBP, RecQL4 and Cdc45 is consistent with the increased origin activation we observed after Rif1 depletion. The increased chromatin binding of Cdc7/Drf1 after Rif1 depletion could be an indirect effect related to Rif1's role in regulating nuclear architecture, as it was suggested in mammalian cells, where DDK accumulation was observed to a lesser extent²² than we show in here in *Xenopus*. Alternatively, it could also be due to a direct inhibition of DDK by Rif1/PP1; in budding yeast, an interaction between Rif1 and Dbf4 has been observed, but the removal of Rif1 did not alter DDK activity^{24,47}. The moderate decrease of TopBP1, also observed in a former study³², could be linked to its replication-independent roles⁴⁸. In the absence of Rif1 and PP1, slower migrating forms of hyperphosphorylated MCM4 were detected, as demonstrated in a former study in *Xenopus*²⁵. We have previously shown that Treslin and MTBP are phosphorylated in a DDK-dependent manner⁴². If Rif1 would recruit PP1 also to pre-initiation complexes, we expected that MTBP/Treslin would remain phosphorylated in the absence of Rif1/PP1. We found that the slower migrating forms of Treslin and MTBP were enriched on chromatin compared to the quicker migrating forms in the absence of Rif1 in Fig. 5b and in Fig. 5d, respectively. Dephosphorylation by λ -phosphatase treatment confirmed that the slower migrating bands after Rif1 depletion represents a phosphorylated form of MTBP or Treslin (Supplementary Fig. 6b). These results show that the phosphorylation state of Treslin and MTBP depends on the presence of Rif1 in *Xenopus*. A similar effect was described in budding yeast for the Treslin homolog Sld3⁴⁷. This observation could be explained by Rif1 recruiting PP1 or another phosphatase on the pre-initiation complex. Alternatively, maybe not exclusive, this could be the

consequence of a Rif1-dependent inhibition of DDK recruitment (or activity), since an enrichment of DDK was observed after Rif1 depletion. The significance of the DDK-dependent phosphorylation and the Rif1-dependent dephosphorylation on Treslin/MTBP activity remains to be explored in future studies; MTBP phosphorylation by CDK or checkpoint kinases regulates origin firing, possibly by controlling the stability of the MTBP-Treslin complex^{12,42,49}. Altogether, our observation of proteins enriched on chromatin after Rif1 depletion provides strong support for the prediction of the numerical model that Rif1 limits initiation factors."

2. Their mathematical model of replication kinetics is a useful tool for developing and testing hypotheses, for instance to test if intuitive models are quantitatively plausible, or to suggest the response of parameters to experimental perturbation that could be experimentally investigated. However, it is not able to robustly predict which parameters change response in a given experimental condition. It is said that with 4 free parameters, one can fit an elephant https://en.wikipedia.org/wiki/Von_Neumann's_elephant.

The authors' model has 7 free parameters. In any case, they are generally appropriately cautious about conclusions drawn from their model. However, they overstate their case, on Page 11, Line 1, where they claim, about their model analysis, that "this demonstrates that Rif1 depletion mainly leads to ...". They should revise the phrase to "this analysis suggests that Rif1 depletion may lead to"

Answer: We agree with the referee that one should not overemphasize a numerical model, and we tried not to do so. We corrected the sentence in the revised version on page 11 and also elsewhere to lower the overstatement (see point 1). Regarding the reviewer's doubts about the number of parameters used to construct the numerical model: The model belongs to a process-based model family, meaning that we were looking for the minimal number of interacting processes necessary to describe the experimental data. We built it by using a nesting procedure that we have previously tested (Ciardo et al. 2021, Genes) by two objective measurements: one statistics; F test, and the other information based; Akaike's criterion. The ability of different models with increasing complexity to describe replication kinetics and the model used in this paper provided the optimized numbers of processes to best describe the experimental data. The seven parameters of the model describe the three considered processes (the bimolecular firing process, the genome segmentation, and the fork-modulated initiation). This does not exclude that other less complex models, not considered in the work can also describe the data but confirms the strength of our model in comparison to the multiple models tested. To comment on the referee's Von Neumann's quotation, "With four parameters I can fit an elephant...": In this affirmation, Von Neumann questions the problem of overfitting; in Ciardo *et al.*, 2021, Genes, we precisely avoid this problem by assessing the fitting ability of our numerical model using the F test and Akaike's criterion, which have demonstrated that the proposed model does not overfit the experimental data (see Supplementary in Ciardo et al., 2021, Genes).

As mentioned by the reviewer, our mathematical model of replication kinetics is indeed "a useful tool to suggest the response of parameters to experimental perturbation that could be experimentally investigated". This is precisely how we used it, and we think combining numerical predictions with experiments strengthens our study. We never claimed our model could "robustly predict which parameters change"; but "its robustness to simulate the replication process under different experimental conditions^{42,43}", which is different to us. We used it to suggest changes that we tried to verify experimentally. Using this strategy, we found that after Rif1 depletion, firstly, the number of foci corresponding to the numerical value θ increased and secondly, known limiting factors corresponding to the numerical value N_0 also increased. The usefulness of this model was also found in another study on the role of Plk1 (Ciardo et al. 2021, NAR).

We now write in the text on page 7, line 19:

"Altogether, this suggests that Rif1 depletion may lead to *i*) an increase in the fraction of regions with highly efficient, early origins (θ), *ii*) an increase in origin firing for late dispersed origins (*Pout*) and *iii*) a small increase of origin firing inside early regions (*Pin*)."

3. Speaking of models, one fits models to data, not vice versa. On Page 9, Line 15 they should replace “data to our” to “data with our”.

Answer: Thank you, we agree and apologize for the mistake. We have corrected on page 7, line 4:

“To provide a more comprehensive picture of the role of Rif1 in DNA replication, we fitted our recent minimal dynamic model of DNA replication, which had proven its robustness to simulate the replication process under different experimental conditions^{42,43}, to our Mock and Rif1 depletion combing data.”

3. The other subject on which the authors should be more measured is the hypothesis that Rif1 excludes limiting replication factors from late replicating chromatin. They provide no evidence that address how Rif1 regulates the function of limiting replication factors, but a simpler model is that it creates a zone of locally high phosphatase activity in regions of late replicating chromatin. On Page 15, Lines 8-9, they present both hypotheses, which is fine. However, in the Abstract, they only present the exclusion hypothesis. They should rephrase the last sentence of the Abstract from "probably by excluding limiting replication factors, such as MTBP, from late replicating chromatin" to "possibly by inactivating limiting replication factors, such as MTBP, in, or excluding them from, regions of late replicating chromatin".

Answer: We agree that both hypotheses (high PP1 concentration and exclusion of initiation factors) could fit our observations, as we wrote on page 14 of the original manuscript. We provided evidence that Rif1/PP1 either directly or indirectly controls the phosphorylation status of MTBP and now also its partner Treslin. In addition, our data in new Fig. 5b, c also show that Cdc7/Drf1 was increased on chromatin after Rif1 depletion. DDK acts in parallel with PP1 and the DDK-dependent phosphorylation of MCM4 has been described as one of the first steps in the cascade of origin activation. Either Rif1/PP1 interacts with DDK to inhibit its chromatin binding, or Rif1 creates a chromatin structure that excludes DDK and other “limiting factors” (MTBP/Treslin, RecQL4) from chromatin. The Rif1-dependent exclusion of DDK from chromatin has already been proposed in a former study (Yamazaki et al. 2012); the authors observed a small and punctual enrichment of DDK in early S phase of mammalian cells in the absence of Rif1. We show here a significant enrichment of Cdc7 and its regulating partner Drf1 though-out S phase in three independent experiments. Rif1 may create a PP1-rich environment, thereby inhibiting MTBP/Treslin and, in parallel, excluding DDK, Treslin/MTBP, and RecQL4 from chromatin. Since MTBP/Treslin are known to be phosphorylated in a DDK-dependent manner, another possibility would be that after Rif1 depletion, more DDK is recruited, leading to an increased Treslin/MTBP on chromatin. We agree that further experiments would be needed to distinguish between these hypotheses.

We rephrased the last sentence of the abstract as suggested page 2 line 12.

“We show that Rif1 globally, but not locally, restrains the replication program in early embryos, possibly by inhibiting or excluding replication factors from chromatin.”

With the new additional data, we added the following text now on pages 8-9, line 18-13:

“To experimentally test our unexpected *in silico* prediction.... Altogether, our analysis of enriched proteins after Rif1 depletion experimentally strengthens the prediction of the numerical model that Rif1 limits initiation factors.”

On page 9, line 29:

“Rather than being concentration limited, we propose that DDK and firing factors might be excluded from chromatin by more Rif1-compacted chromatin domains, or their activity might be regulated at some origins or regions by Rif1-dependent PP1 targeting. Another possibility could be that self-organizing Rif1-PP1 hubs break down when DDK/CDK levels reach a threshold level leading to a more or less synchronized origin firing in later replicating sequences as proposed for satellite sequences in *Drosophila*²⁸.”

And on page 10, line 13:

“This could be by directly inhibiting initiation factors like Treslin/MTBP and/or indirectly by modulating DDK access to chromatin and the ability of MCM2-7 to compete for limiting factors. In the absence of Rif1, structurally more relaxed domains²³ and the absence of PP1 would allow the binding of DDK kinase and firing factors, such as Treslin/MTBP and RecQL4, to origins and would result in a boost of initiations in whole clusters, as we observed by DNA combing analysis.”

Reviewer #3 (Remarks to the Author):

General comments

The major claim of this paper is that Rif1 depletion strongly accelerates DNA synthesis in egg extracts by advancing the activation of replication clusters at the beginning of the S phase, whereas it only modestly increases origin activation inside clusters and increases the apparent fork speed. Authors also show that the phosphorylation of MTBP1 is accelerated in Rif1 depleted extracts.

Since the discovery of Rif1 as a crucial regulator of replication timing, a number of reports have examined the effects of Rif1 on replication timing profiles. It has been well established that the Rif1 affects the profile of replication timing domains; in most cases, midS-late replication domains are converted to early replicating domains, indicating replication timing changes in a domain-specific manners (corresponding to the “clusters of the origins discussed in this report). Therefore, it may not be surprising that the replication timing changes in "clusters". However, the precise mode of origin regulation by Rif1 in early embryos have not been known.

Authors presented data showing that Rif1 increases the “early clusters” but not the firing frequency of individual origins and does not affect the fork rate.

Combing analyses have been professionally conducted and the data of combing and its analyses are convincing. The data provide conclusive information that in the *Xenopus* extracts, firing efficiency and fork rate is increased, contrary to the expectation from some of the data from yeasts and mammals. The data are important in that it illustrates the role of Rif1 in regulation of DNA replication in early embryos.

Suggestions for improvement

Authors presented data showing that Rif1 increases the “early clusters” but not the firing frequency of individual origins and does not affect the fork rate.

The data of combing and its analyses are convincing but there is no information regarding the clusters.

1. How are they distributed along the genome, and how are they affected by Rif1 depletion. How many clusters on the genome and what portions of the “late-firing clusters” are converted to early firing clusters?

Answer: We cannot determine by a DNA combing experiment how clusters are distributed along the genome since our experimental approach does not provide information on their genomic localization. We visualize origin activation on fibers without knowing to which regions in the genome they belong. In our system with a rapid S phase, cluster size, number of origins per cluster, as well as their time of activation varies, which makes a direct approach for the cluster analysis challenging. In the original manuscript in Fig. 2, we showed that fork density and initiation frequency increased without Rif1, whereas origin distances (ETED) remained mainly unchanged locally. This discrepancy is due to the fact that origins are organized in clusters, as described before by us. In addition, we showed in Fig. 2c of the original manuscript that after Rif1 depletion, the number of non-replicated fibers strongly decreased, consistent with more activated clusters. To better describe the effect of Rif1 depletion on

cluster organization, we now looked in more detail at the distribution of replication tracks at the level of individual fibers (as already described in Marheineke&Hyrien, JBC, 2004), which indirectly shows differences and lets us estimate changes in cluster activation after Rif1 depletion. We added three more graphs in the revised version of the manuscript. First, we verified in new Supplementary Fig. 2a, b that the number of replication tracks per fiber is not randomly distributed in mock or Rif depleted conditions. We observe an excess of fibers with either no or more than three or five replication tracks, respectively, compared to a random theoretical distribution (same comparison as in Marheineke&Hyrien, JBC 2004). Second, we replaced Fig. 2c with a new Fig. 2d, which now shows the distribution of the percentage of fibers exhibiting an increasing number of replication tracks per fiber in mock and Rif1 depleted conditions for early time points representing early S phase (50, 60, 75 min). To perform this analysis, we chose a fiber subpopulation (n=2519 fibers) with an identical fiber size distribution to avoid a bias since fibers from Rif1 depleted conditions were slightly smaller. New Fig. 2d shows fewer unreplicated fibers after Rif1 depletion. In addition, we observe also 5 to 10 fold more fibers with more than 5 replication tracks per fiber after Rif1 depletion. This suggests that either, with constant intra-cluster origin distances, clusters contain more origins or several more clusters are activated in the absence of Rif1. Third, as requested by the referee, we estimated the number of activated clusters after Rif1 depletion early in S phase and compared it to the control. To do this, we calculated the excluded ETEDs, as described in Marheineke and Hyrien 2004, and compared Rif1-depleted to mock-depleted extracts in the New Supplementary Fig. 3b and the modified Supplementary Table. Eye-to-eye-distance measurement requires the presence of at least 2 eyes on a fiber meaning 2 individualized tracks. Therefore, fibers with limited lengths, incomplete eyes or gaps are normally excluded from ETED measurements; this implies that large inter-cluster ETEDs have a higher probability to be excluded. The excluded ETED value takes into account this omission, as does the fork density value also. The value of the excluded ETEDs reflects the inter-cluster distance, as explained in our 2004 study. This allowed us to compare two experimental conditions by calculating the ratio of excluded ETED. We found that the mean excluded ETEDs ranged from 5 to 1.2 fold greater in controls compared to the Rif1 depletion according to S phase progression, indicating that more clusters are activated in the absence of Rif1. Based on these values, in the absence of Rif1, we estimate a 5-fold increase of clusters activated during the early S phase going down to 2-fold during the mid S phase.

To be more explicit, we also changed the order of our rationales; we now show the local ETEDs distribution on Fig. 2c before the N of tracks/fiber distribution in Fig. 2d.

The changes within the text are as follows on pages 5-6 line 30-22.

“We calculated initiation frequencies (number of initiations per time unit per unit length of unreplicated DNA), $I(t)$, and found that initiations were about three-fold higher during the early S phase after Rif1 depletion compared to the control versus a two-fold increase at mid-S phase (Fig. 2b). Thus, Rif1 depletion led to a substantial increase in initiations, especially during the early S phase. We, therefore, expected to observe a sharp decrease of origin distances. However, we found that measured origin distances (eye-to-eye distances, ETEDs) remained unchanged during the early S phase and only moderately decreased in mid S phase (Fig. 2c, median decrease 1.1-1.4 fold) in the absence of Rif1. The apparent discrepancy between the global initiation frequency and ETEDs measured on single fibers can be explained by the limit in fiber length set by DNA breaks or microscope field and the organization of 2 to 8 origins per cluster^{5,6}. To study differences in cluster activation in both conditions, we analyzed the distribution of all replication tracks per fiber as previously described⁶. Both distributions contained an excess of fibers with either no eye or multiple eyes compared to a random distribution (Supplementary Fig. 2a, b), consistent with the fact that origins are not activated independently of each other but in clusters. Moreover, the distributions from Mock and Rif1 depleted extracts differed significantly (Fig. 2d; p-value = $4.77 \cdot 10^{-32}$, Mann-Whitney test). After Rif1 depletion, we observed that the percentage of fibers without replication tracks was reduced during the early S phase. Conversely, the percentage of fibers containing more than 2 labeled replication tracks per fiber sharply increased. Importantly, fibers showing more than 5 tracks were 5 to 10 fold more frequent, suggesting either larger clusters or more activated clusters in the absence of Rif1 (representative fibers in Supplementary Fig. 3a). Given that incomplete eyes or gaps are excluded from ETED measured on fibers (or intra-cluster

ETED), it follows that large inter-cluster ETEDs have a higher probability of being excluded. We thus also analyzed these excluded ETED (or inter-cluster ETED) for both conditions (Supplementary Fig. 3b, Supplementary Table), using a method previously described⁶. We observed that these distances were 5.3 to 1.2 fold larger from the very early to the mid S phase, respectively, in controls compared to Rif1-depleted extracts. This reflects that, depending on S phase progression, 5.3 to 1.2 more replication clusters were activated in the absence of Rif1. The observed changes in the replication track patterns, the inter-cluster distances (excluded distances), together with the moderate changes in the intra-cluster ETED, suggest that the initiation increase after Rif1 depletion was mainly caused by the activation of whole replication clusters.”

2. What are the features of the affected clusters (Rif1 binding, epigenetic feature, transcription activities and so on). Some of these questions need to be addressed.

Answer: It would be very interesting to access the characteristics of the clusters affected by Rif1 depletion. As discussed above, we cannot tell which features clusters are associated with throughout the genome. For this, it would be necessary to carry out a genomic approach using NGS and Repli-Seq analysis, which was not the aim of our study and which is actually challenging due to the poor quality of the *Xenopus laevis* genome assembly. Even if we had genomic data, we could not link changes with transcription because, as mentioned in the manuscript (page 3, line 8), there is no transcription in early embryos and in the *Xenopus in vitro* system, transcription only starts after the MBT, after 12 cell cycles. The chromatin structure is not yet shaped as in differentiated cells and partitioned into eu- and heterochromatin. Using immunofluorescence, in Fig. 1d, e, we show that Rif1 binds rather homogeneously during early S phase and co-localizes with late replicating regions (page 5).

3. In authors previous publications, it was stated that the each *Xenopus* cluster contain 5– 8 origins and span 50–100 kb. The current DNA fibers would contain one or two clusters. If longer DNA fibers containing many more clusters can be visualized and compared between the mock and Rif1-depleted extracts, it could be more visually convincing and would provide additional information on the nature of origin clusters and its regulation by Rif1.

Answer: We now show selected long fibers (around 260 kb) from mock and Rif1 depleted conditions to representatively illustrate conclusions from the cluster organization (analysed in Fig. 2 d) in a new Supplementary Fig. 3a. After Rif1 depletion, we often find fibers with many tracks on. We also present one very long specific fiber (from 2 consecutive fields) of over 330 kb that clearly shows two clusters, which have been activated at different times (left: 3 separate tracks and right: one large track which has arisen from fusion of several tracks activated before the 3 left tracks). We found only very few long fibers in our experiments due to frequent breaks during genomic DNA preparation despite careful handling. Therefore, statistics on very long fibers is challenging.

We write in the text on page 6 line 11

"Importantly, fibers showing more than 5 tracks were 5 to 10 fold more frequent, suggesting either larger clusters or more activated clusters in the absence of Rif1 (representative fibers in Supplementary Fig. 3a)."

4. Concern on the conclusion

Authors conclude that Rif1 binding to mid-late regions or domains restricts the accessibility of initiation factors to chromatin by keeping close domain contacts and establishing locally high phosphatase activity, which slows down S phase progression. The model is related to what was proposed before and can explain the data presented. However, in the absence of the knowledge on the nature of the origin clusters and Rif1 binding profile in the egg extracts, Rif1 could be simply inhibiting the PP1 that counteracts the phosphorylation on preRC factors mediated by DDK, and thus regulating the association of initiation factors including MTBP1.

Answer: We agree that the conclusion that Rif1 depletion affects late clusters is based on the kinetics of our synchronous systems (higher effect early S phase, lower effect mid S phase), our immunofluorescence data (Rif1 localization to later replicating regions) and the literature. Since there is little effect after Rif1 depletion on origin distances, this argues for a more global impact of Rif1, not on the level of single origins but on the level of clusters. Chip-Seq and Repli-Seq would be most interesting to perform in this system, but as explained above, these experimental approaches are very challenging.

As discussed on page 15 of the original manuscript, the two models are not mutually exclusive and could fit our explanations. On one hand, Rif1 could increase PP1 inside clusters or domains; on the other hand, it could exclude initiation factors. Our new data now also show an enrichment of DDK and RecQL4 on chromatin after Rif1 depletion, reinforcing the “exclusion hypothesis”.

We corrected in the abstract, page 2 line 12:

“We show that Rif1 globally, but not locally, restrains the replication program in early embryos, possibly by inhibiting or excluding replication factors from chromatin.”

On page 9, line 29, we introduced the following text:

“Rather than being concentration limited, we propose that DDK and origin firing factors might be excluded from chromatin by more Rif1-compacted chromatin domains, or their activity might be regulated at some origins or regions by Rif1-dependent PP1 targeting. Another possibility could be that self-organizing Rif1-PP1 hubs break down when DDK/CDK levels reach a threshold level leading to a more or less synchronized origin firing in later replicating sequences as proposed for satellite sequences in *Drosophila*²⁸.”

On page 10, line 13

“This could be by directly inhibiting initiation factors like Treslin/MTBP and/or indirectly by modulating DDK access to chromatin and the ability of MCM2-7 to compete for limiting factors. In the absence of Rif1, structurally more relaxed domains²³ and the absence of PP1 would allow the binding of DDK kinase and firing factors, such as Treslin/MTBP and RecQL4, to origins and would result in a boost of initiations in entire clusters, as we observed by DNA combing analysis.”

Other concerns

5. The MTBP data in Figure 5 appears to be preliminary. There is no information on the phosphorylation induced by Rif1 depletion,

Answer: We showed in three independent experiments that the phosphorylation and MTBP amount increase after Rif1 depletion. Phosphatase treatment demonstrates that mobility shifts are linked to phosphorylations. We do not know yet which residues inside MTBP are phosphorylated and dephosphorylated in this DDK/Rif1-dependent manner. This would be very interesting to learn and could be the subject of future studies. We found 21 potential DDK phosphorylation sites in MTBP.

or on how the time course of chromatin association of MTBP (in the absence of Rif1) is related to that of other factors (DDK, RecQL4, MCM10, Cdc45, TopBP1 and others).

Answer: We thank the referee for this thoughtful suggestion, which we followed up. Our new results in new Fig. 5b, c now strengthen the fact that Rif1 depletion increases the availability of limiting factors. We now show western blots for Cdc7 kinase and its regulatory subunit Drf1 (embryonic Dbf4 homolog),

Treslin, TopBP1 and RecQL4. We found that the binding of DDK, Treslin and RecQL4 to chromatin, like that of MTBP, is significantly increased at the start of the kinetics and throughout S phase whereas the increase in Cdc45 is observed only later, consistent with an increase of origin firing. The increase of these factors have been quantified in three independent experiments and is significant. The increase of MTBP/Treslin and RecQL4, as well as the strong increase of DDK on chromatin after Rif1 has never been described before in *Xenopus* or other systems and are therefore very intriguing. DDK (Drf1), Treslin, RecQL4, and TopBP1 are the limiting replication factors described in yeast and *Xenopus* (Mantiero et al, 2011, Collart et al., 2013). We did not detect any increase of TopBP1 recruitment but a slight significant decrease that could reflect other functions of TopBP1 besides its role in DNA replication after Rif1 depletion. A former study also described this decrease (Kumar et al., 2012). Altogether, these new results support the predictions of the numerical model ascribing a role for Rif1 in limiting initiation factors, either by direct inhibition or indirectly by excluding them through the structural role of Rif1 in the nucleus. The MCM10 protein has not been described as rate limiting and is also recruited later in the cascade. We considered it less relevant since we designed our study to check-out the prediction of our mathematical model for increasing limiting factors. We modified Fig. 5 by adding these new western blots in Fig. 5b and in new Supplementary Fig. 5a, quantifications obtained from three replicates are presented in Fig. 5c. The phosphorylation of MTBP is now shown in Fig. 5d.

We modified the text as follows, page 8-9, lines 17-13:

“It was uncertain whether Rif1 could regulate limiting initiation factors, essential for maturing the pre-replication complexes into the pre-initiation complexes in vertebrates^{11,14}. To experimentally test our unexpected *in silico* prediction, we analyzed by western blotting the effect of Rif1 depletion on the binding to chromatin of several origin firing factors (Treslin, MTBP, TopBP1, RecQL4) and the S phase kinase Cdc7/Drf1 (DDK). After Rif1 depletion, the total amount of chromatin-bound Treslin, MTBP, Cdc7, Drf1, RecQL4, and Cdc45 significantly increased in three replicates, whereas TopBP1 slightly decreased throughout the S phase (Fig. 5b-c, Supplementary Fig. 6a). The increase of the pre-IC proteins Treslin/MTBP, RecQL4 and Cdc45 is consistent with the increased origin activation we observed after Rif1 depletion. The increased chromatin binding of Cdc7/Drf1 after Rif1 depletion could be an indirect effect related to Rif1's role in regulating nuclear architecture, as it was suggested in mammalian cells, where DDK accumulation was observed to a lesser extent²² than we show in here in *Xenopus*. Alternatively, it could also be due to a direct inhibition of DDK by Rif1/PP1; in budding yeast, an interaction between Rif1 and Dbf4 has been observed, but the removal of Rif1 did not alter DDK activity^{24,47}. The moderate decrease of TopBP1, also observed in a former study³², could be linked to its replication-independent roles⁴⁸. In the absence of Rif1 and PP1, slower migrating forms of hyperphosphorylated MCM4 were detected, as demonstrated in a former study in *Xenopus*²⁵..... Altogether, our analysis of enriched proteins after Rif1 depletion provides strong support for the prediction of the numerical model that Rif1 limits initiation factors.”

We also modified the model in new Fig. 6 accordingly to introduce DDK, Treslin and RecQL4 and revised the conclusions on page 9 and 10:

Page 9, line 26:

“Second, origin firing factors such as MTBP/Treslin, Drf1, and RecQL4 do not seem to be concentration limiting *per se* in *Xenopus* egg extracts since, in the absence of Rif1, more firing factors and Drf1 are recruited to chromatin, and many more origins can be activated. Rather than being concentration limited, we propose that DDK and firing factors might be excluded from chromatin by more Rif1-compacted chromatin domains, or their activity might be regulated at some origins or regions by Rif1-dependent PP1 targeting.”

Page 10, line 10:

“We propose a model for Rif1 function in early embryos where Rif1 binding to mid-late regions or domains restricts the accessibility of initiation factors to chromatin by keeping close domain contacts²³

and by establishing locally high phosphatase activity, which slows down S phase progression (Fig. 6). This could be by directly inhibiting initiation factors like Treslin/MTBP and/or indirectly by modulating DDK access to chromatin and the ability of MCM2-7 to compete for limiting factors. In the absence of Rif1, structurally more relaxed domains²³ and the absence of PP1 would allow the binding of DDK kinase and firing factors, such as Treslin/MTBP and RecQL4, to origins and would result in a boost of initiations in entire clusters, as we observed by DNA combing analysis.”

6. There is no add-back experiment of Rif1. Authors cannot completely rule out the possibility that immunodepletion depleted unknown Rif1 associated proteins, and would be more reassuring if one data for the recovery experiment could be presented.

Answer: We agree with the referee that an add-back experiment would have been an additional supportive experiment. However, Rif1 is a very large protein (257 kDa) lacking well-defined functional domains and is predicted to be largely intrinsically disordered; these features have hampered the recombinant expression of Rif1 and subsequent functional characterization (Sukackaite et al., 2014). We were not able to produce the full-length, recombinant, functional Rif1 protein. To our knowledge, no other *Xenopus* lab has so far reported this and thus provided an add-back experiment after Rif1 depletion (Kumar et al., 2012, Alver et al., 2017). Whereas the smaller, full-length recombinant *S.cerevisiae* Rif1 protein (MW 218KDa) could have been obtained (Kanoh et al., 2015), the overexpressed vertebrate forms seem to be more challenging to purify after overexpression. The murine Rif1 protein appeared unstable and was rapidly degraded after homologous overexpression and purification (Moriyama et al., 2018). Therefore, most studies used truncations of this large proteins to study the different domains in biochemical assays. Add-back experiments in *Xenopus* egg extracts ensure that what is observed is not the indirect effect of potential co-depletions. We cannot completely exclude that some unknown proteins are co-depleted with Rif1 from egg extracts. However, no co-depletion of checkpoint proteins like TopBP1, ATR, or ATM has been observed (Kumar et al., 2012). In addition, two different proteomic approaches to identify Rif interactors in Mice and *Xenopus* did not reveal other known DNA replication initiation inhibitors than PP1.

We propose to add these explanations in the text page 9 line 13:

“We were not able to generate a recombinant form of Rif1 that could rescue the depleted extracts. However, vertebrate Rif1 is a very large protein and is difficult to produce in an active form in recombinant systems^{50,51}. Immunodepletion of Rif1 does not noticeably decrease the abundance of TopBP1 and other known checkpoint regulators in egg extracts³², and two different proteomics approaches in *Xenopus* and mouse have not revealed Rif1 interactions with known negative replication regulators other than PP1^{25,52}.”

7. Minor points

Page 15, line 8; Rif -> Rif1

Answer: Thank you, this has been changed.

Other changes made by the authors:

We also shortened the abstract as recommended by guidelines of the journal.

In Figure 4: the complete data set has been used now to generate new graphs, with no impact on the shape of the curves.

In Figure 6, we combined former panels **a** and **b** to present one more concise model and added DDK, Treslin and RecQL4.

Modified Figures:

We updated **Fig. 2**, where panel d was moved up to c, and a new panel d was created (answering referee 3 concerning the cluster analysis)

Fig. 2: Rif1 depletion strongly increases origin activation in the early-S phase at the level of replication clusters.

a Principle of DNA combing experiment with a fiber example and measured parameters. Sperm is replicated in control (ΔMock) or Rif1 immunodepleted (ΔRif1) egg extract in the presence of biotin dUTP, DNA was isolated at the indicated times, and then DNA combing was performed in two independent experiments. **b** Mean initiation frequencies ($I(t)$) with standard error of mean (SEM) and ratio $\Delta\text{Rif1}/\Delta\text{Mock}$ ($I(t)$) were calculated. **c** Scatter dot plot of eye-to-eye distances (ETED) at different time points from replicate 1, median with interquartile range, Mann-Whitney test. **d** Percentage of unreplicated fibers and fibers with increasing number of replication tracks per fiber from early S phase from both independent experiments, with an equivalent fiber length distribution in mock and Rif1 depleted condition; percentage ratios indicated below each class. ($n = 2519$ fibers for each distribution, Mann-Whitney test on distribution, p -value 4×10^{-41}). **e** Scatter dot plot of eye lengths (EL), replicate 1, median with interquartile range, Mann-Whitney test; n indicated below, ns: non-significant, * indicates significant difference (Mann-Whitney U test, two-sided, $p < 0.05$; p -values: * 0.01-0.05; ** 0.001-0.01; *** 0.0001-0.001; **** < 0.0001).

We changed **Fig. 5 b-d**; it now shows 6 more chromatin-bound proteins (Treslin, Cdc7, Drf1, RecQL4, Cdc45, TopBP1) and their changes after Rif1 depletion on panel **b**, panel **c** shows their quantification. Unchanged panel **b** moved to **d** (answering to referee 1,2,3). Panel **a** remains unchanged

Fig. 5: Rif1 depletion increases the chromatin recruitment of initiation factors.

a Inferred model parameters by fitting Δ Mock and Δ Rif1 depleted combing data of two replicates at a comparable percentage of replication. Circles indicate the mean value of the parameters over 100 different runs of the genetic algorithm; the error bars are standard deviations, (*) indicates a significant difference between the Δ Mock and Δ Rif1 samples, χ^2 test as described⁴². **b** Chromatin fractions after Rif1 or Mock depletion were isolated at indicated times and analyzed by western blot for indicated proteins. **c** Ratios of Δ Rif/ Δ Mock optical densities (OD) of western blot bands for indicated proteins from three independent chromatin experiments, normalized to Orc2-OD, scatter dot plot with mean and SD, including all time points ($n = 10-15$), one sample t-test. **d** Phosphorylation of MTBP after Rif1 depletion, chromatin proteins were separated after 60 min using Anderson-SDS-PAGE to better resolve phosphorylated forms of MTBP⁴³.

We changed **Fig. 6**, to simplify the model and include new data (from **Fig. 5 b,c**).

Fig. 6: Model for Rif1 restraining the spatio-temporal replication program in early *Xenopus* embryos.

Rif1 orchestrates the temporal availability of initiation factors and PP1 in early *Xenopus* embryo's DNA replication at the level of domains; for details, see text.

New Supplementary Figures:

We inserted a new **Supplementary Figure 2**, to show referee 3 the number of labeled track per fiber for mock and Rif1 depleted condition compared to a random distribution.

Supplementary Figure 2: Non-homogenous organisation of replication tracks on combed fibers after mock and Rif1 depletion.

Using the dataset from Fig. 2d, frequency of combed fibers versus the number of replication tracks per fiber (N) for early S phase time points (exp) compared to a Poisson distribution (random) with identical mean (theo). **a** From mock depleted extracts, Chi-squared test ($p = 1.4 \times 10^{-164}$). **b** From Rif1 depleted extracts, Chi-squared test, $p = 0$.

We inserted a new **Supplementary Figure 3**, to answer referee 3. Example of fibers on panel **a** and excluded ETEDs values that allow us to estimate the fold change of cluster activation on panel **b**.

Supplementary Figure 3: Replication track patterns differ in the presence and absence of Rif1.

a Example of combed fibers with DNA (green) and replication eyes (red) from Fig. 2d, early time points. Positions of replication tracks indicated below each fiber in bold black. Third fiber in Rif1 depleted conditions is a composed fiber of two consecutive microscope fields. **b** Mean excluded eye-to-eye distances (mean excluded ETED) from mock and Rif1 depleted extracts of experiment1; corresponding ratios are indicated below.

Mean excluded ETED = excluded DNA / ((N of forks/2) - N of ETED);
 excluded DNA = tot DNA - (mean ETED x N of ETED).

Old **Supplementary Figure 2** becomes **Supplementary Figure 4**

Old **Supplementary Figure 3** becomes **Supplementary Figure 5**

We replaced **old Supplementary Figure 4** by the **Supplementary Figure 6**: to answer referees 1, 2, and 3, we added a phosphatase treatment of Treslin and show a biological replicate of the blot shown on **Fig. 5b**.

Supplementary Figure 6: Rif1 depletion increases limiting replication factors and p-MTBP, p-Treslin on chromatin.

a Chromatin fractions after Rif1 depletion or Mock depletion were isolated at different times and analyzed by western blot for indicated proteins, second biological replicate of time course experiment as in Fig. 5. **b** samples from Fig. 5b, treated with λ -phosphatase.

Old **Supplementary Figure 5** becomes **Supplementary Figure 7**

REVIEWERS' COMMENTS:

Reviewer #2 (Remarks to the Author):

The authors have satisfactorily addressed my concerns.

Reviewer #3 (Remarks to the Author):

In the revised manuscript by Haccard, the authors responded to the comments from me and from other reviewers with additional analyses of data and additional experiments. The authors responses are mostly satisfactory and I am happy with the revised manuscript.

It was nice to see new data in Figure 5bc that show the absence of Rif1 increases the chromatin association of initiation factors, although the exact mechanism by which Rif1 excludes (or limit the amount of) these factors from chromatin will need to be answered by future studies.

The authors did not conduct the ChIP seq analyses of the Rif1 binding sites in *Xenopus* egg extracts, which would be very interesting, and also did not conduct the add back experiments. I understand the technical difficulties associated with these experiments, and agree that the current manuscript can be accepted for publication in *Communications Biology* without these experiments.

REVIEWERS' COMMENTS:

Reviewer #2 (Remarks to the Author):

The authors have satisfactorily addressed my concerns.

Reviewer #3 (Remarks to the Author):

In the revised manuscript by Haccard, the authors responded to the comments from me and from other reviewers with additional analyses of data and additional experiments. The authors responses are mostly satisfactory and I am happy with the revised manuscript.

It was nice to see new data in Figure 5bc that show the absence of Rif1 increases the chromatin association of initiation factors, although the exact mechanism by which Rif1 excludes (or limit the amount of) these factors from chromatin will need to be answered by future studies.

The authors did not conduct the ChIP seq analyses of the Rif1 binding sites in *Xenopus* egg extracts, which would be very interesting, and also did not conduct the add back experiments. I understand the technical difficulties associated with these experiments, and agree that the current manuscript can be accepted for publication in *Communications Biology* without these experiments.

Answer: We would like to thank all the reviewers for their valuable and constructive comments and for understanding our experimental system.